# T-cell receptor (TCR) signaling promotes the assembly of RanBP2/RanGAP1-SUMO1/Ubc9 nuclear pore subcomplex via PKC-θ-mediated phosphorylation of RanGAP1

Yujiao He[1], Zhiguo Yang[1†], Chen-si Zhao[1†], Zhihui Xiao[1†], Yu Gong[1], Yun-Yi Li[1], Yiqi Chen[1], Yunting Du[1], Dianying Feng[1], Amnon Altman[2], Yingqiu Li[1]*

[1]MOE Key Laboratory of Gene Function and Regulation, Guangdong Province Key Laboratory of Pharmaceutical Functional Genes, State Key Laboratory of Biocontrol, School of Life Sciences, Sun Yat-sen University, Guangzhou, China; [2]Center for Cancer Immunotherapy, La Jolla Institute for Immunology, La Jolla, United States

**Abstract** The nuclear pore complex (NPC) is the sole and selective gateway for nuclear transport, and its dysfunction has been associated with many diseases. The metazoan NPC subcomplex RanBP2, which consists of RanBP2 (Nup358), RanGAP1-SUMO1, and Ubc9, regulates the assembly and function of the NPC. The roles of immune signaling in regulation of NPC remain poorly understood. Here, we show that in human and murine T cells, following T-cell receptor (TCR) stimulation, protein kinase C-θ (PKC-θ) directly phosphorylates RanGAP1 to facilitate RanBP2 subcomplex assembly and nuclear import and, thus, the nuclear translocation of AP-1 transcription factor. Mechanistically, TCR stimulation induces the translocation of activated PKC-θ to the NPC, where it interacts with and phosphorylates RanGAP1 on Ser[504] and Ser[506]. RanGAP1 phosphorylation increases its binding affinity for Ubc9, thereby promoting sumoylation of RanGAP1 and, finally, assembly of the RanBP2 subcomplex. Our findings reveal an unexpected role of PKC-θ as a direct regulator of nuclear import and uncover a phosphorylation-dependent sumoylation of RanGAP1, delineating a novel link between TCR signaling and assembly of the RanBP2 NPC subcomplex.

*For correspondence:
lsslyq@mail.sysu.edu.cn

[†]These authors contributed equally to this work

Competing interests: The authors declare that no competing interests exist.

## Introduction

Nuclear pore complexes (NPCs) span the nuclear envelope (NE) and mediate nucleo-cytoplasmic exchange. NPC dysfunction has been associated with various human diseases (*Beck and Hurt, 2017*). NPCs in the animal kingdom are ~110 megadalton supramolecular assemblies of multiple copies of ~30 different nuclear pore proteins, termed nucleoporins (NUPs) (*Hampoelz et al., 2019a*; *Lin and Hoelz, 2019*). The elaborate structure of NPCs consists of several biochemically and ultra-structurally defined substructures, namely, cytoplasmic filaments, cytoplasmic and nuclear rings, the inner pore ring, the central transporter region, and the nuclear basket (*Hampoelz et al., 2019a*; *Lin and Hoelz, 2019*; *Otsuka and Ellenberg, 2018*).

An important component of the cytoplasmic filaments is the RanBP2 subcomplex that consists of RanBP2 (Ran-binding protein 2, Nup358), RanGAP1 (Ran GTPase-activating protein 1) SUMO1, and Ubc9 (SUMO-conjugating enzyme) (*Hampoelz et al., 2019a*; *Lin and Hoelz, 2019*; *Werner et al., 2012*). This subcomplex has multiple functions in nucleo-cytoplasmic transport by establishing a RanGTP gradient across the NE and coordinating recycling of importin-β and reassembly of novel

import complexes to promote importin-α/importin-β heterodimer-mediated nuclear import (*Hamada et al., 2011*; *Hampoelz et al., 2019a*; *Hutten et al., 2008*; *Lin and Hoelz, 2019*), serving as a disassembly machine for CRM1-dependent nuclear export complexes (*Ritterhoff et al., 2016*), stabilizing the interaction between the inner and outer Y complex of the cytoplasmic ring in humans (*von Appen et al., 2015*), and controlling NPC assembly beyond nuclei during oogenesis (*Hampoelz et al., 2019b*). RanGAP1 has two forms, i.e., non-sumoylated RanGAP1 that mainly localizes in the cytoplasm, and SUMO1-conjugated RanGAP1 that resides at NPCs (*Matunis et al., 1996*), generated by Ubc9-mediated sumoylation (*Bernier-Villamor et al., 2002*; *Lee et al., 1998*). SUMO1 conjugation of RanGAP1 is required for assembly of the RanBP2 subcomplex (*Hampoelz et al., 2019a*; *Hampoelz et al., 2019b*; *Hutten et al., 2008*; *Joseph et al., 2004*; *Mahajan et al., 1997*; *Reverter and Lima, 2005*; *Ritterhoff et al., 2016*; *von Appen et al., 2015*; *Werner et al., 2012*). However, it is unclear whether receptor signaling and, particularly, immune cell stimuli, regulate RanBP2 subcomplex assembly.

Protein kinase C-θ (PKC-θ) is a pivotal regulator of T-cell activation. PKC-θ belongs to the novel, calcium-independent subfamily of the PKC enzyme family and is highly expressed in hematopoietic cells, particularly in T cells (*Baier et al., 1993*). Engagement of the T-cell receptor (TCR) together with costimulatory receptors (e.g., CD28) by cognate antigens and costimulatory ligands presented by antigen-presenting cells (APCs) recruits PKC-θ to the center of the T-cell immunological synapse (IS) formed at the T cell–APC contact site, where it mediates activation of the transcription factors nuclear factor kappa B (NF-κB), AP-1, and nuclear factor of activated T-cells (NFAT), leading to T-cell activation, cytokine (e.g., IL-2) production, and acquisition of effector functions (*Baier-Bitterlich et al., 1996*; *Coudronniere et al., 2000*; *Li et al., 2004*; *Lin et al., 2000*; *Pfeifhofer et al., 2003*; *Wang et al., 2015*; *Xie et al., 2019*).

AP-1 transcription factors have pleiotropic effects, including in different aspects of the immune system such as T-cell activation, Th differentiation, T-cell anergy, and exhaustion (*Atsaves et al., 2019*). To date, unlike the regulation of NF-κB, which has been studied extensively, the regulation of AP-1 by PKC-θ in response to TCR stimulation is incompletely understood. Paradoxically, although PKC-θ deficiency does not obviously impair the TCR-induced activation of c-Jun N-terminal kinase (JNK) nor the expression level of AP-1 in mature T cells, it severely inhibits AP-1 transcriptional activity (*Pfeifhofer et al., 2003*; *Sun et al., 2000*), suggesting the existence of an unknown regulatory layer of TCR-PKC-θ signaling. Hence, further exploring how PKC-θ mediates TCR-induced AP-1 activation is important for understanding T-cell immunity.

In this study, we reveal that PKC-θ directly regulates the nuclear import function of the NPC, which accounts for the effective, TCR-induced activation of AP-1. We show that PKC-θ promotes the nuclear import process by facilitating the assembly of the RanBP2 subcomplex in T cells. This nuclear import was significantly impaired in PKC-θ-deficient ($Prkcq^{-/-}$ or KO) T cells. Mechanistically, PKC-θ directly interacted with and phosphorylated RanGAP1 at $Ser^{504}$ and $Ser^{506}$. RanGAP1 phosphorylation at these two Ser residues, but not sumoylation at $Lys^{524}$, promotes its binding to Ubc9, thereby facilitating the sumoylation of RanGAP1 and, consequently, RanBP2 subcomplex assembly. We further demonstrate that a non-phosphorylatable double mutant (S504A/S506A; AA) of RanGAP1 displayed a significantly reduced sumoylation and blocked TCR-induced nuclear translocation of AP-1, as well as that of NF-κB and NFAT. A RanGAP1-EE mutant (S504E/S506E; EE) that mimics phosphorylation of RanGAP1 displayed an increased sumoylation, and its forced expression rescued TCR-induced c-Jun nuclear translocation in PKC-θ knockdown T cells. Thus, our study demonstrates a novel PKC-θ-mediated, phosphorylation-dependent sumoylation of RanGAP1 as an obligatory step for proper assembly of the RanBP2 subcomplex and, furthermore, establishes a novel mechanism for the regulation of AP-1, NF-κB, and NFAT activation by PKC-θ.

## Results

### PKC-*θ* translocates to the NE and colocalizes with NPCs upon TCR stimulation

Several PKC isoforms translocate to the NE after phorbol ester (PMA, a diacylglycerol mimetic and PKC activator) treatment (*Collas, 1999*; *Leach et al., 1989*). We therefore examined whether PKC-θ also translocates to the NE in stimulated Jurkat E6.1 T cells, a human leukemic T-cell line widely

used for the study of TCR signaling (*Abraham and Weiss, 2004*). By biochemically isolating the NE, we observed that anti-CD3 plus anti-CD28 antibody (Ab) costimulation or stimulation with APCs that were pulsed with a superantigen, staphylococcal enterotoxin E (SEE), promoted PKC-θ translocation to the NE fraction (*Figure 1A,B*, respectively). Consistent with previous findings, TCR stimulation also promoted PKC-θ translocation to the plasma membrane (PM) fraction (*Figure 1A,B*). In addition, transmission electron microscopy analysis of immunogold-labeled PKC-θ showed its translocation to the cytoplasmic face of the NE in response to anti-CD3 plus anti-CD28 costimulation (*Figure 1—figure supplement 1A*). Quantitation showed that after 15 min of stimulation, >20% of total PKC-θ was localized 100 nm or less from the NE and >30% of total PKC-θ was localized 100 nm or less from the PM (*Figure 1—figure supplement 1B*).

Given that NPCs are the important components of the NE, we next determined whether PKC-θ colocalized with NPCs. We stimulated Jurkat T cells with anti-CD3 plus anti-CD28 Abs or with PMA plus a $Ca^{2+}$ ionophore, ionomycin (Iono), for 15 min and imaged the cells by confocal microscopy. Staining with an anti-PKC-θ Ab and a monoclonal Ab (Mab414) that recognizes a subset of NPC proteins revealed that stimulation significantly increased PKC-θ colocalization with NPCs (*Figure 1C,D*). Similarly, T cells stimulated with SEE-pulsed APCs also displayed partial PKC-θ colocalization with the NPCs (*Figure 1—figure supplement 1C*). Immunoprecipitation (IP) with Mab414 showed that costimulation increased PKC-θ binding to NPCs in primary human T cells (*Figure 1E*) and in Jurkat T cells (*Figure 1F*). When we stimulated T cells with PMA plus Iono, a portion of PKC-θ molecules translocated to the NE (*Figure 1—figure supplement 1D-F*). Together, these results demonstrate that TCR stimulation induces PKC-θ translocation to the NE and, more specifically, colocalization and physical association with NPCs, suggesting that PKC-θ may participate in the process of nucleo-cytoplasmic transport.

## PKC-*θ* deficiency decreases NPC association of importin-β1

Importin-β1 and Ran proteins play a key role in nucleo-cytoplasmic transport, and importin-β1 can be specifically recruited to NPCs and then mediate the passage of proteins, including transcription factors such as AP-1, NF-κB, and NFAT, into the nucleus through NPCs (*Tetenbaum-Novatt and Rout, 2010*; *van der Watt et al., 2016*). To determine the physiological relevance of the PKC-θ–NPC association, we first analyzed the cellular distribution of importin-β1 and Ran in resting T cells from wild-type and $Prkcq^{-/-}$ mice. Using confocal microscopy, we found that PKC-θ deletion resulted in a decreased ratio of nuclear-to-cytoplasmic of importin-β1 and Ran (*Figure 1G*, *Figure 1—figure supplement 1G*). PKC-θ deletion also resulted in a decreased translocation of importin-β1 and Ran to the nuclear fraction (*Figure 1H*, *Figure 1—figure supplement 1H*). Next, we assessed the binding of importin-β1 to NPCs. In wild-type T cells, importin-β1 constitutively coimmunoprecipitated with NPCs, and anti-CD3 plus anti-CD28 costimulation significantly increased their association; in contrast, in $Prkcq^{-/-}$ T cells, importin-β1 barely bound to NPCs, regardless of stimulation (*Figure 1I*, *Figure 1—figure supplement 1I*). Similar results were found when we knocked down PKC-θ expression in Jurkat T cells with a specific small interfering RNA or short hairpin RNA (siPKC-θ or shPKC-θ, respectively) (*Figure 1J–L*, *Figure 1—figure supplement 1J-M*). These data indicate that PKC-θ deficiency alters both the basal state and TCR-induced nucleo-cytoplasmic transport.

## PKC-*θ* binds to RanGAP1 at the NE in a sumoylation-dependent manner

To further investigate how PKC-θ regulates nuclear transport, we constructed GST-tagged proteins, including nucleoporins, RanGAP1, and other NE proteins, and performed GST pull-down assays to determine whether these proteins can interact with PKC-θ. Among them, GST-RanGAP1 showed the most obvious association and a direct interaction with PKC-θ in unstimulated Jurkat cells lysate (*Figure 2A,B*, *Figure 2—figure supplement 1A*). Reciprocal co-IP from Jurkat T cells showed that PKC-θ weakly interacted constitutively with sumoylated RanGAP1, and costimulation with anti-CD3 plus anti-CD28 Abs markedly increased the association of PKC-θ with both forms of RanGAP1 (*Figure 2C*). A significant colocalization of PKC-θ and RanGAP1 in NPCs was also observed in Jurkat T cells after anti-CD3 plus anti-CD28 or PMA plus Iono stimulation (*Figure 2—figure supplement 1B,C*).

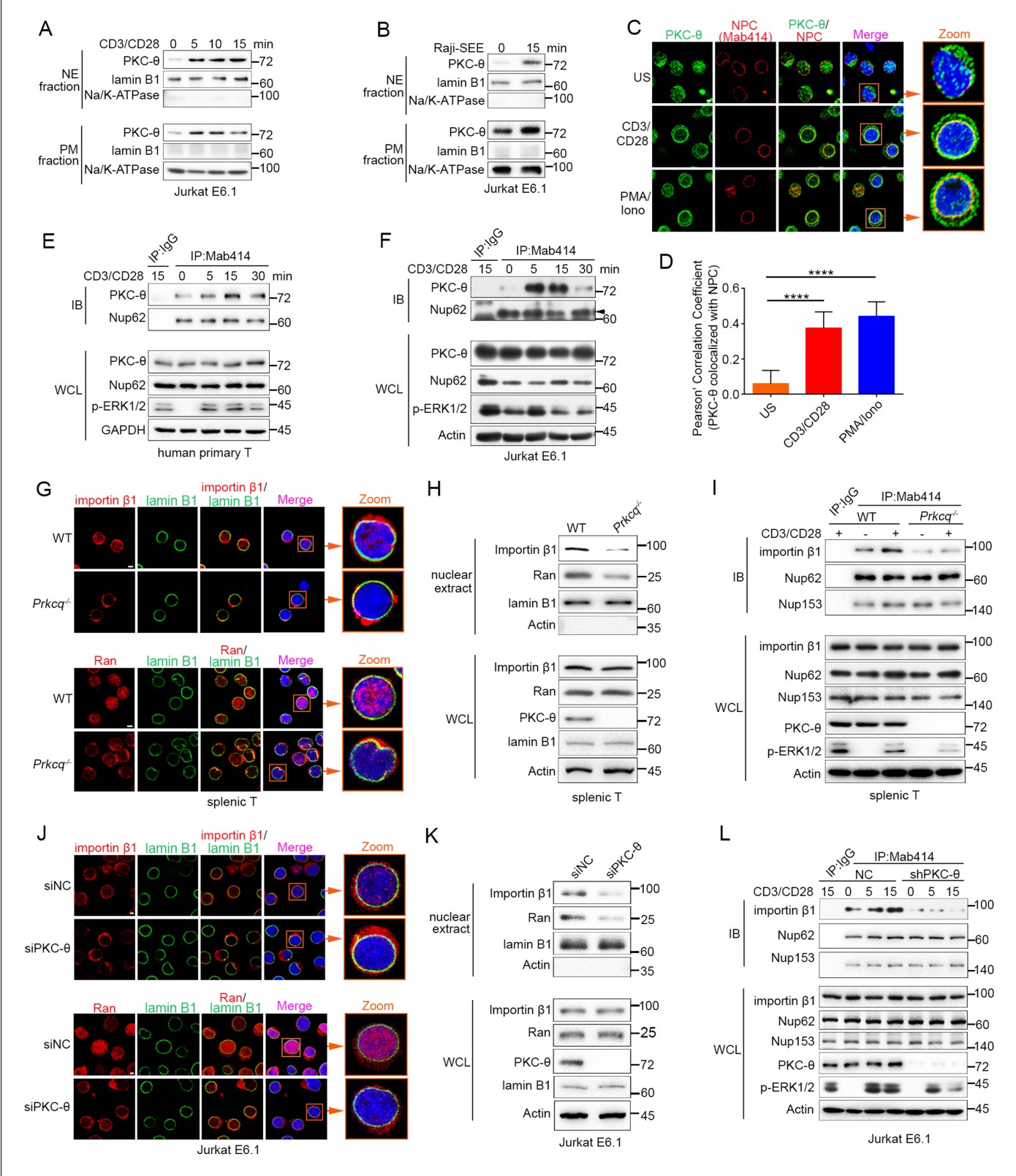

**Figure 1.** TCR stimulation promotes PKC-θ colocalization with the NPC and PKC-θ deficiency decreases nuclear import of importin-β1 and Ran and NPC association with importin-β1. (**A, B**) Subcellular fractionation of Jurkat E6.1 cells stimulated for 0–15 min with anti-CD3 plus anti-CD28 (**A**) or with superantigen (SEE)-pulsed Raji B cells (**B**) and immunodetection with the indicated antibodies. NE, nuclear envelope; PM, plasma membrane. (**C**) Confocal imaging of PKC-θ (green) and NPCs (Mab414, red) colocalization in representative Jurkat E6.1 cells left unstimulated (US) or stimulated for 15

*Figure 1 continued on next page*

*Figure 1 continued*

min with anti-CD3 plus anti-CD28 or with PMA plus Iono. Nuclei are stained with DAPI (blue). Areas outlined by squares in the merged images are enlarged at right. Scale bars, 2 µm. (**D**) Quantification of PKC-θ colocalization with NPCs by Pearson correlation coefficient. Analysis was based on at least three different images covering dozens of cells using the ImageJ software. ****p<0.0001 (one-way ANOVA with post hoc test). (**E, F**) Immunoblot analysis of NPC IPs (Mab414) or whole-cell lysates (WCL) from human primary T cells (**E**) or Jurkat E6.1 cells (**F**) stimulated for 0–30 min with anti-CD3 plus anti-CD28. Control IP with normal IgG is shown in the left lane. Nup62, an NPC component, was used a as loading control for the IPs. The arrowhead indicates the Nup62 protein band. (**G**) Confocal imaging of importin-β1 and Ran in representative wild-type (WT) or *Prkcq*⁻/⁻ mouse primary splenic T cells stained with the indicated antibodies. Areas outlined by squares in the merged images are enlarged at right. Scale bars, 2 µm. (**H**) Subcellular fractionation of mouse splenic T cells and immunodetection with the indicated antibodies. (**I**) Immunoblot analysis of NPC IPs (Mab414) or whole-cell lysates (WCL) from unstimulated or anti-CD3 plus anti-CD28-stimulated WT or *Prkcq*⁻/⁻ mouse splenic T cells. Control IP with normal IgG is shown in the left lane. (**J, K**) Confocal imaging of importin-β1 and Ran (**J**) and subcellular fractionation (**K**), analyzed as in (**G, H**), of Jurkat E6.1 cells transfected with scrambled siRNA-negative control (siNC) or PKC-θ targeting siRNA (siPKC-θ). Scale bars, 2 µm. (**L**) Immunoblot analysis of NPC IPs (Mab414) or WCL from unstimulated or stimulated Jurkat E6.1 T cells stably expressing a control small hairpin RNA (shRNA) or a PKC-θ targeting shRNA (shPKC-θ), analyzed as in (**I**). Data are representative of three (**A, B, E, F, H, I, K, L**) or two (**C, D, G, J**) biological replicates.

The online version of this article includes the following source data and figure supplement(s) for figure 1:

**Source data 1.** Uncropped western blot for *Figure 1*.
**Source data 2.** Row data for *Figure 1* and for *Figure 1—figure supplement 1*.
**Figure supplement 1.** PKC-θ translocates to the NE following TCR stimulation and PKC-θ deficiency decreases nuclear import of importin β1 and Ran and NPC association with importin β1.
**Figure supplement 1—source data 1.** Uncropped western blot for *Figure 1—figure supplement 1*.

To elucidate the contribution of sumoylation to the association of RanGAP1 with PKC-θ, we transfected Jurkat T cells with wild-type or non-sumoylated RanGAP1 mutant (RanGAP1-K524R) having an N-terminal Flag tag or a C-terminal HA tag. PKC-θ coimmunoprecipitated efficiently with Ran-GAP1-K524R (*Figure 2D*, *Figure 2—figure supplement 1D*), implying that RanGAP1 sumoylation is not required for its interaction with PKC-θ. Next, we mapped the PKC-θ determinants required for its NE translocation and RanGAP1 interaction by transfecting Jurkat T cells with c-Myc-tagged wild-type PKC-θ, a desumoylated mutant PKC-θ−2KR (K325R/K506R) (*Wang et al., 2015*), a constitutively active PKC-θ-A148E mutant, or a catalytically inactive PKC-θ-K409R mutant. Reciprocal IP showed that both the sumoylated and non-sumoylated forms of RanGAP1 interacted more strongly with PKC-θ-A148E than with wild-type PKC-θ, whereas PKC-θ−2KR and PKC-θ-K409R displayed a much weaker interaction, if at all (*Figure 2E*, *Figure 2—figure supplement 1E,F*). Following biochemical isolation of the NE, we observed that wild-type PKC-θ translocated to the NE in response to PMA plus Iono stimulation (*Figure 2F*, *Figure 2—figure supplement 1G*), consistent with the results in *Figure 1*. Interestingly, PKC-θ-A148E was constitutively localized to the NE regardless of stimulation, while no apparent NE localization of either PKC-θ−2KR or PKC-θ-K409R was observed even after PMA plus Iono stimulation (*Figure 2F*, *Figure 2—figure supplement 1G*), consistent with the result in *Figure 2E*.

We next compared the sumoylation of wild-type PKC-θ and its mutants. As expected, wild-type PKC-θ, but not PKC-θ−2KR, was sumoylated (*Figure 2G*). Interestingly, PKC-θ-K409R, like PKC-θ−2KR, could not be sumoylated, whereas PKC-θ-A148E was sumoylated more strongly than wild-type PKC-θ (*Figure 2G*, *Figure 2—figure supplement 1H*), indicating that the catalytic activity of PKC-θ is required for its sumoylation. Combining the results in *Figure 2E–G* with the result in *Figure 2D*, we conclude that PKC-θ sumoylation, rather than the sumoylation of RanGAP1, was important for their association and that PKC-θ sumoylation is required for its NE translocation.

## PKC-θ deficiency inhibits the association of RanGAP1 with the NPC by reducing its sumoylation

Given the association between PKC-θ and RanGAP1, we next explored the physiological significance of this interaction. We first inspected the localization of RanGAP1 in resting T cells from wild-type or *Prkcq*⁻/⁻ mice. Confocal microscopy revealed that NE-localized RanGAP1 was decreased in *Prkcq*⁻/⁻ T cells, with most RanGAP1 found in the cytosol (*Figure 3A*). When we used siPKC-θ to transiently knock down endogenous PKC-θ in Jurkat T cells, the amount of cytosol-localized RanGAP1 increased, while NE-localized RanGAP1 decreased (*Figure 3B*). Consistent with this result, stimulation of human primary T cells with anti-CD3 plus anti-CD28 or with PMA plus Iono increased the ratio

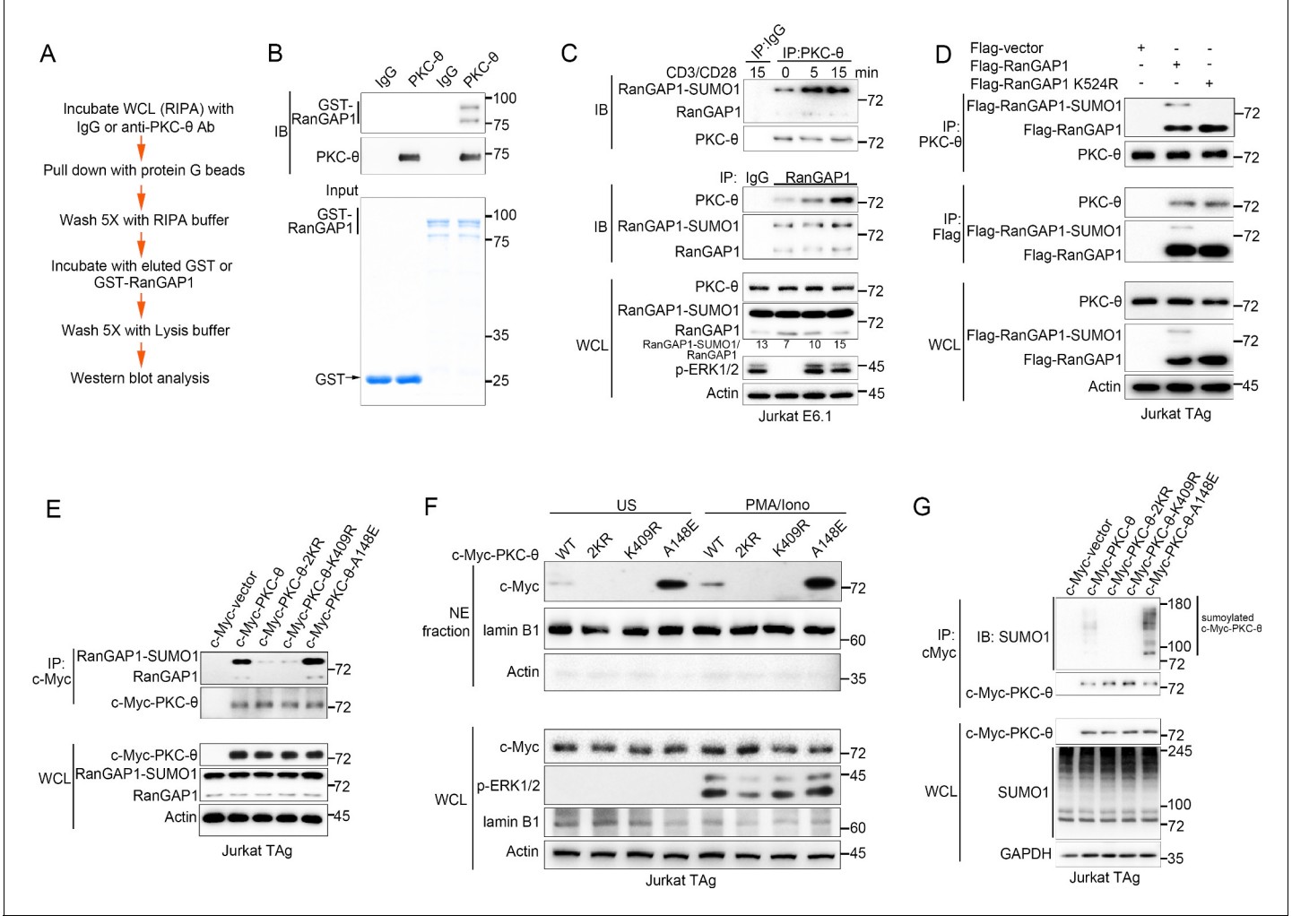

**Figure 2.** PKC-θ association with both RanGAP1 and RanGAP1-SUMO1 requires PKC-θ sumoylation. (**A**) Scheme for in vitro protein direct binding assay. (**B**) Analysis of direct association between RanGAP1 and PKC-θ using approach in (**A**). GST-fusion proteins were detected by Coomassie blue staining (bottom). (**C**) Immunoblot analysis of PKC-θ or RanGAP1 IPs or WCL from Jurkat E6.1 cells stimulated for 0–15 min with anti-CD3 plus anti-CD28. (**D**) Reciprocal IP analysis of the association between endogenous PKC-θ and transfected Flag-tagged wild-type or mutated RanGAP1 in Jurkat-TAg cells. (**E**) Immunoblot analysis of c-Myc-tagged PKC-θ IPs or WCL from Jurkat-TAg cells that were transiently transfected with wild-type PKC-θ or the indicated PKC-θ mutants. (**F**) Subcellular fractionation of Jurkat-TAg cells transiently transfected with wild-type PKC-θ or the indicated PKC-θ mutants (2KR, K325R/K506R desumoylation mutant; K409R, kinase dead; A148E, constitutive active), followed by immunodetection with the indicated antibodies. (**G**) Immunoblot analysis of c-Myc-tagged PKC-θ IPs or WCL from Jurkat-TAg cells that were transiently transfected with the indicated expression vectors. Data are representative of at least three biological replicates (**B–G**).

The online version of this article includes the following source data and figure supplement(s) for figure 2:

**Source data 1.** Uncropped western blot for *Figure 2*.
**Figure supplement 1.** PKC-θ binds to and colocalizes with RanGAP1 in NE.
**Figure supplement 1—source data 1.** Uncropped western blot for *Figure 2—figure supplement 1*.
**Figure supplement 1—source data 2.** Row data for *Figure 2—figure supplement 1*.

of sumoylated to unsumoylated RanGAP1 (*Figure 3C*). A similar increase in this ratio was observed when mouse primary T cells or Jurkat T cells were costimulated with anti-CD3 plus anti-CD28 Abs; however, this increase was not observed when expression of PKC-θ was knocked out or knocked down (*Figure 3D,E*, *Figure 3—figure supplement 1A*, respectively; whole-cell lysates [WCL]). Consistently, PKC-θ deficiency also decreased the ratio in resting cells (*Figure 3D,E*, respectively; WCL and *Figure 3—figure supplement 1B*).

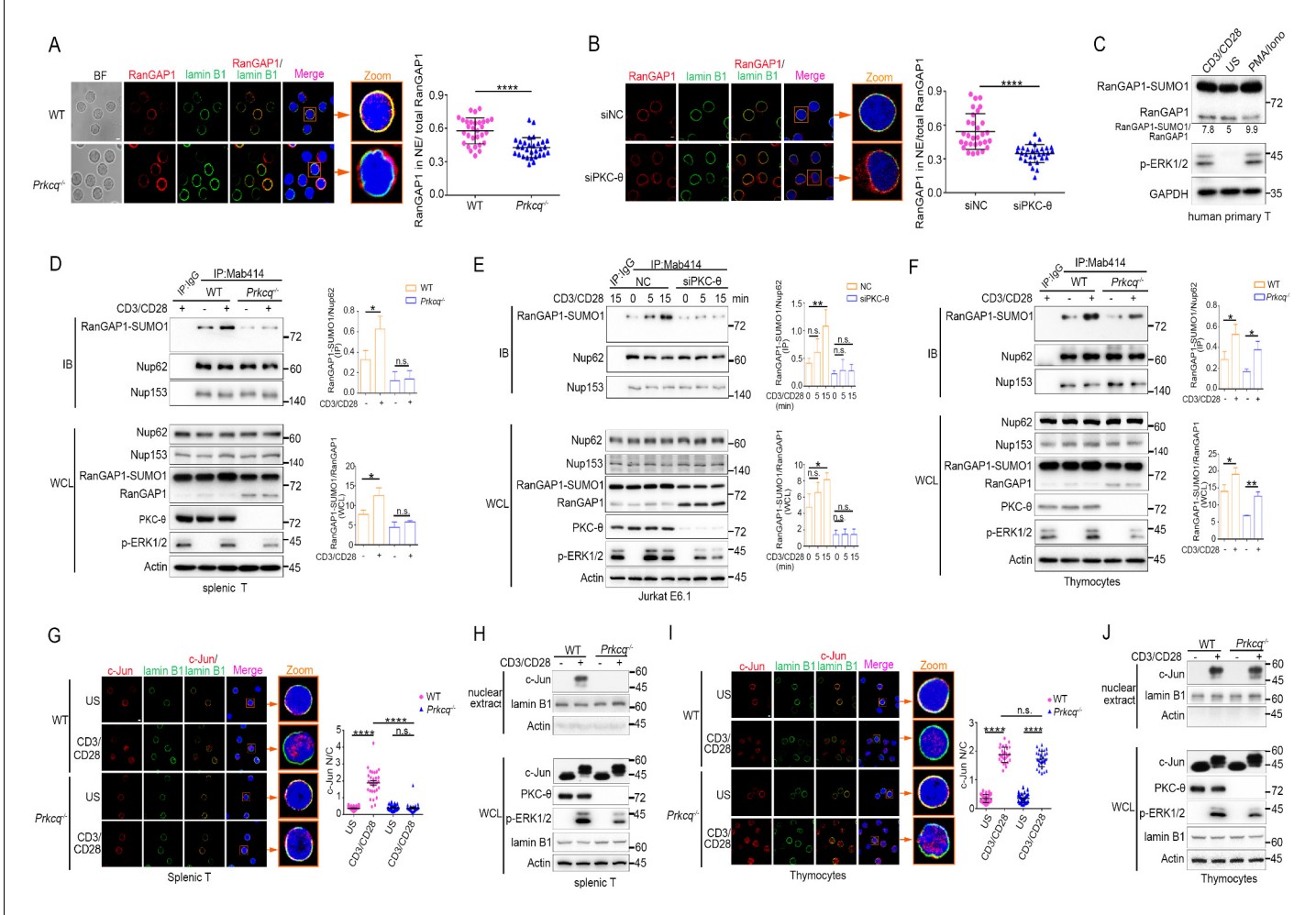

**Figure 3.** PKC-θ deficiency inhibits RanGAP1 sumoylation and its incorporation into the NPC and TCR-induced nuclear import of c-Jun in mature T cells. (A, B) Confocal imaging of RanGAP1 (red) and lamin B1 (green) localization in representative unstimulated WT or *Prkcq*[−/−] mouse primary T cells (A), or in representative unstimulated Jurkat E6.1 cells transfected with siNC or siPKC-θ (B). Nuclei are stained with DAPI (blue). Areas outlined by squares in the merged images from (A) or (B) are enlarged at right in (A) or (B). Scale bars, 2 μm. Quantification of the ratio of NE/total RanGAP1 based on analysis of ~30 cells in about six random fields from two biological replicates as presented in (A) or (B) is shown at far right in (A) or (B), with each symbol representing an individual cell. Horizontal lines indicate the mean ± s.e.m. ****p<0.0001 (two-tailed, unpaired Student's t-test). (C) Immunoblot analysis of RanGAP1-SUMO1 and RanGAP1 in human primary T cells unstimulated or stimulated for 15 min with anti-CD3 plus anti-CD28 or PMA plus Iono. (D–F) Immunoblot analysis of Mab414 IPs or WCL from WT and *Prkcq*[−/−] mouse primary splenic T cells (D) and thymocytes (F) stimulated with or without anti-CD3 plus anti-CD28 for 15 min, or from Jurkat E6.1 cells transfected with siNC or siPKC-θ stimulated for 0–15 min with anti-CD3 plus anti-CD28. (E) Statistical analysis of the amount ratio of RanGAP1-SUMO1 to Nup62 or to RanGAP1 in IPs or in WCL from the experiment in (D), (E), or (F) is shown at right in each figure. Analysis was based on three biological replicates for each experiment. n.s., not significant, *p<0.05, **p<0.01 (one-way ANOVA with post hoc test). (G, I) Confocal imaging of c-Jun (red) localization in representative WT or *Prkcq*[−/−] mouse splenic T cells (G) or thymocytes (I) unstimulated (US) or stimulated with anti-CD3 plus anti-CD28, costained with anti-lamin B1 (green) and DAPI (blue). Areas outlined by squares in the merged images from (G) or (I) are enlarged at right in (G) or (I). Scale bars, 2 μm. Quantitative analysis of the N/C ratio of c-Jun analyzed in ~30 cells in about six random fields from two biological replicates as presented in in (G) or (I) is shown at far right in (G) or (I). Each symbol represents an individual cell. Horizontal lines indicate the mean ± s.e.m. n.s., not significant; ****p<0.0001 (one-way ANOVA with post hoc test). (H, J) Subcellular fractionation of WT or *Prkcq*[−/−] mouse splenic T cells (H) or thymocytes (J) stimulated for 0–15 min with anti-CD3 plus anti-CD28, followed by immunodetection with the indicated antibodies. Data are representative of two (A, B, G, I) or three (C–F, H, J) biological replicates.

The online version of this article includes the following source data and figure supplement(s) for figure 3:

**Source data 1.** Uncropped western blot for *Figure 3*.

**Source data 2.** Row data for *Figure 3* and for *Figure 3—figure supplement 1*.

**Figure supplement 1.** PKC-θ deficiency inhibits RanGAP1 sumoylation and its incorporation into the NPC, but does not affect TCR-induced phosphorylation of c-Jun.

**Figure supplement 1—source data 1.** Uncropped western blot for *Figure 3—figure supplement 1*.

Sumoylated RanGAP1 was present in Mab414 NPC IPs prepared from mouse primary peripheral T cells or Jurkat T cells regardless of stimulation, but was largely absent or substantially reduced when these cells were depleted of PKC-θ both in resting and in activated T cells (*Figure 3D,E*, *Figure 3—figure supplement 1A*, respectively; IP and *Figure 3—figure supplement 1B*). Together, these results indicate that PKC-θ promotes the sumoylation of RanGAP1 as well as the incorporation of RanGAP1 into the NPC both in resting and activated cells. Interestingly, however, and in contrast to peripheral T cells or Jurkat cells, in *Prkcq*$^{-/-}$ mouse thymocytes, TCR plus CD28 costimulation appeared to increase the association of RanGAP1 with the NPC (*Figure 3F*; IP) as well as its relative increased sumoylation (*Figure 3F*; WCL) in a manner that was not affected by PKC-θ deletion, although PKC-θ deficiency did decrease the ratio of RanGAP1-SUMO1 to RanGAP1 in resting thymocytes (*Figure 3F*, WCL), similar to the mature T cells (*Figure 3D*, WCL). Upon TCR stimulation, ERK phosphorylation was decreased in *Prkcq*$^{-/-}$ mature T cells (*Figure 3D*, WCL), consistent with the finding that PKC-θ activates ERK via RasGRP1-Ras signaling (*Roose et al., 2005*). Notably, the activation of ERK was also reduced in *Prkcq*$^{-/-}$ thymocytes (*Figure 3F*, WCL), consistent with a previous report (*Morley et al., 2008*), but in contrast to another report (*Pfeifhofer et al., 2003*; *Sun et al., 2000*). These apparently contrasting results about the effect of *Prkcq* deletion on ERK activation in thymocytes could be due to the fact that the earlier study *Pfeifhofer et al., 2003*; *Sun et al., 2000* used *Prkcq*$^{-/-}$ mice on a mixed background, whereas the later study (*Morley et al., 2008*) and ours used mice that were extensively backcrossed on the B6 background. Collectively, these results indicate that PKC-θ controls constitutive RanGAP1 sumoylation in both resting mature and immature T cells, but is critical for the TCR-induced RanGAP1 sumoylation and RanGAP1-SUMO association with NPCs only in mature T cells.

## PKC-θ deficiency inhibits TCR-induced nuclear import of c-Jun in mature, but not immature T cells

Given the finding that PKC-θ is required for AP-1 activation in mature T cells, but not in thymocytes (*Sun et al., 2000*) and the difference between thymocytes and peripheral T cells observed above (*Figure 3D* vs. *Figure 3F*), we considered the possibility that the activation and/or nuclear import of c-Jun, which is a component of AP-1 transcription factor, may display difference in its PKC-θ dependence in mature peripheral T cells vs. thymocytes. Whereas PKC-θ deficiency did not affect TCR-induced phosphorylation of JNK and c-Jun (indicative of its activation) in either mouse splenic T cells or thymocytes (*Figure 3—figure supplement 1C,D*), confocal microscopy and nuclear fractionation showed that the TCR-induced nuclear localization of c-Jun was blocked in *Prkcq*$^{-/-}$ splenic T cells (*Figure 3G,H*, *Figure 3—figure supplement 1E*), but remained intact in *Prkcq*$^{-/-}$ thymocytes (*Figure 3I,J*, *Figure 3—figure supplement 1F*), suggesting that PKC-θ regulates the nuclear import of c-Jun through the NPC in mature T cells, but not in thymocytes. To test whether PKC-θ deletion impact is long lasting, we stimulated shPKC-θ cells with anti-CD3 plus anti-CD28 for 0–12 hr. As shown in *Figure 3—figure supplement 1G,H*, after 12 hr stimulation, PKC-θ deletion-induced defects in RanGAP1 sumoylation and its binding to NPCs and c-Jun nuclear translocation still remain. Thus, PKC-θ may promote c-Jun nuclear import, but not its phosphorylation per se, suggesting that it regulates AP-1 activation primarily by controlling the function of the NPC.

## PKC-θ-mediated phosphorylation of RanGAP1 on Ser$^{504}$ and Ser$^{506}$ facilitates its sumoylation

Given the fact that phosphorylation of proteins often regulates their sumoylation (*Hendriks et al., 2017*; *Hietakangas et al., 2006*; *Tomasi and Ramani, 2018*), we next explored the possibility that PKC-θ may regulate the sumoylation of RanGAP1 via phosphorylating it. Using a mixture of phospho-Ser- and phospho-Thr-specific antibodies, we found that TCR plus CD28 costimulation increased phosphorylation of RanGAP1 in control Jurkat T cells, but not in PKC-θ knockdown cells (*Figure 4A*, *Figure 4—figure supplement 1A*). Furthermore, an in vitro kinase assay demonstrated that PKC-θ immunoprecipitated from Jurkat T cells efficiently phosphorylated RanGAP1 (*Figure 4B*), indicating that RanGAP1 is most likely a direct PKC-θ substrate. Mass spectrometry analysis of PKC-θ-phosphorylated RanGAP1 identified five potential phosphorylation serine or threonine sites on RanGAP1 (*Figure 4—figure supplement 1B*). Upon mutating each of these residues individually to alanine, we found that mutation both Ser$^{504}$ and Ser$^{506}$, but not the three other residues, reduced

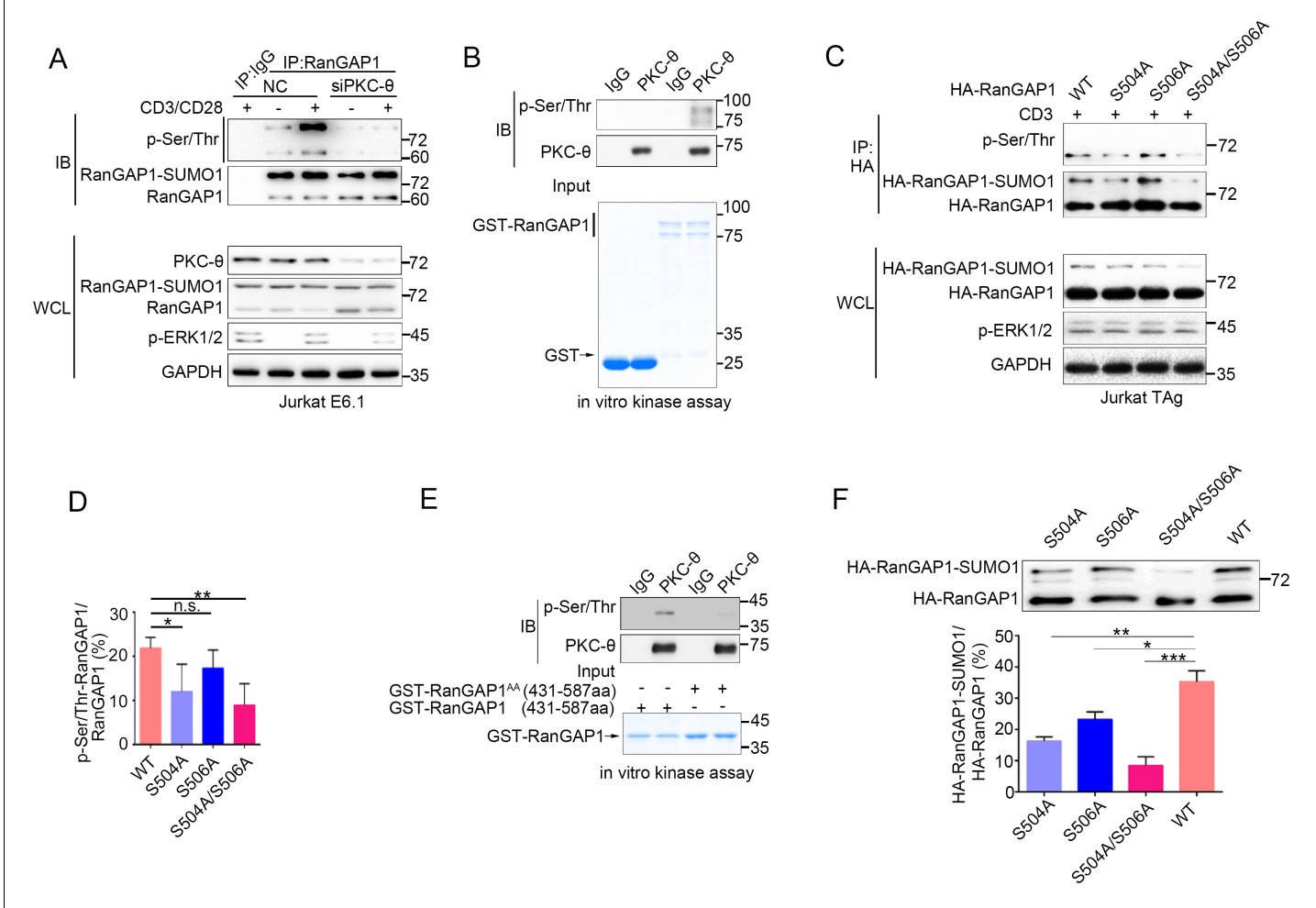

**Figure 4.** PKC-θ-mediated sumoylation of RanGAP1 requires its phosphorylation on Ser[504] and Ser[506]. (**A**) Immunoblot analysis of serine-phosphorylated RanGAP1 in Jurkat E6.1 T cells transfected with siNC or siPKC-θ, and left unstimulated or stimulated with anti-CD3 plus anti-CD28. (**B**) In vitro PKC-θ kinase assay using PKC-θ immunoprecipitated from Jurkat E6.1 cells and recombinant GST-RanGAP1 as substrate. (**C, D**) Immunoblot analysis of the phosphorylation of transfected wild-type or Ser-mutated HA-tagged RanGAP1 using a mixture of p-Ser- and p-Thr-specific antibodies in Jurkat-TAg cells, which were left unstimulated or stimulated for 15 min with anti-CD3 (**C**). Immunoblotting of the indicated proteins in WCL is shown at bottom. The ratio of phospho-RanGAP1 to immunoprecipitated RanGAP1 is shown in (**D**). Analysis is based on three biological replicates. n.s., not significant, *p<0.05, **p<0.01 (one-way ANOVA with post hoc test). (**E**) In vitro PKC-θ kinase assay as in (**B**), using purified truncated GST-RanGAP1 or GST-RanGAP1 [AA] (S504A/S506A) as substrate. (**F**) Immunoblot analysis of the sumoylation of wild-type or the indicated HA-tagged RanGAP1 mutants in Jurkat-TAg cells using an anti-HA antibody (top panel). The ratios of RanGAP1-SUMO1 to RanGAP1 are shown at the bottom panel. Analysis is based on three biological replicates. n.s., not significant, *p<0.05, **p<0.01, ***p<0.001 (one-way ANOVA with post hoc test). Data are representative of three biological replicates.

The online version of this article includes the following source data and figure supplement(s) for figure 4:

**Source data 1.** Uncropped western blot for *Figure 4*.

**Source data 2.** Row data for *Figure 4* and for *Figure 4—figure supplement 1*.

**Figure supplement 1.** PKC-θ-dependent phosphorylation of RanGAP1.

**Figure supplement 1—source data 1.** Uncropped western blot for *Figure 4—figure supplement 1*.

the phosphorylation of RanGAP1, while phosphorylation of the double mutant S504A/S506A (RanGAP1[AA]) was significantly decreased (*Figure 4C,D*, *Figure 4—figure supplement 1C*). In the in vitro kinase assay, we detected phosphorylation of truncated GST-RanGAP1$_{431-587}$ but not the S504A/S506A double mutant (RanGAP1[AA]; *Figure 4E*). Sequence alignment of RanGAP1 from different species showed that S[504] (10/11) and S[506] (8/11) sites and three adjacent serines are conserved (*Figure 4—figure supplement 1D*). A mouse RanGAP1 fragment containing these conserved sites was

also phosphorylated by PKC-θ in an in vitro kinase assay (*Figure 4—figure supplement 1E*). When we calculated the ratio of RanGAP1-SUMO1 to RanGAP1 in the mutant cells, we observed that this ratio was significantly decreased in cells expressing the S504A mutant or the double mutant RanGAP1$^{AA}$ and, to a lesser extent, in S506A-expressing cells (*Figure 4F*). Similar result was also observed when RanGAP1 antibody instead of HA antibody was used to detect HA-RanGAP1 and its SA mutants (*Figure 4—figure supplement 1F*). And HA-RanGAP1 and its SA mutants had a much higher expression level than endogenous RanGAP1, which could be caused by a strong promoter of the expression vector and contributed to the low sumoylation ratio of HA-RanGAP1. Together, these findings suggest that PKC-θ-mediated phosphorylation of RanGAP1, particularly on Ser$^{504}$, promotes its sumoylation.

## PKC-*θ*-mediated phosphorylation of RanGAP1 is required for RanBP2/RanGAP1-SUMO1/Ubc9 subcomplex assembly

The SUMO E2 enzyme Ubc9 directly interacts with and conjugates SUMO1 to RanGAP1 (*Bernier-Villamor et al., 2002*), and the sumoylation of RanGAP1 is essential for assembly of the RanBP2/RanGAP1-SUMO1/Ubc9 subcomplex (*Hampoelz et al., 2019a*; *Hampoelz et al., 2019b*; *Hutten et al., 2008*; *Joseph et al., 2004*; *Mahajan et al., 1997*; *Reverter and Lima, 2005*; *Ritterhoff et al., 2016*; *von Appen et al., 2015*; *Werner et al., 2012*). We therefore hypothesized that PKC-θ-mediated phosphorylation of RanGAP1 might regulate the interaction between Ubc9 and RanGAP1 and, furthermore, that RanGAP1 phosphorylation would be required for assembly of this subcomplex. Hence, we first examined whether RanGAP1 sumoylation affects its binding to Ubc9 and found that a non-sumoylated RanGAP1 mutant (K524R) was still capable of associating with Ubc9 (*Figure 5A*). Next, we generated two RanGAP1 mutants: One, which was mutated at both its sumoylation (K524R) and phosphorylation (S504A/S506A) sites (HA-RanGAP1$^{AA}$/K524R), and another K524R mutant with a replacement of Ser$^{504}$ and Ser$^{506}$ PKC-θ phosphorylation sites by a glutamic acid as a phosphorylation mimic (HA-RanGAP1$^{EE}$/K524R). We then analyzed the association of these mutants with Ubc9 by reciprocal co-IP and found that, compared with HA-RanGAP1-K524R, HA-RanGAP1$^{AA}$-K524R showed a decreased association with Ubc9, while HA-RanGAP1$^{EE}$-K524R had a stronger interaction (*Figure 5A,B*). This result indicates that RanGAP1 phosphorylation promotes its binding to Ubc9 and provides an explanation for our finding that mutation of the RanGAP1 PKC-θ phosphorylation sites inhibits its sumoylation (*Figure 4F*).

Next, we transfected Jurkat T cells with HA-RanGAP1, HA-RanGAP1$^{AA}$, or HA-RanGAP1$^{EE}$ and analyzed their association with Ubc9 and RanBP2 by reciprocal co-IP. HA-RanGAP1$^{AA}$ bound to Ubc9 and RanBP2 less effectively than non-mutated HA-RanGAP1; in contrast, HA-RanGAP1$^{EE}$ bound more effectively to Ubc9 and RanBP2 (*Figure 5C,D*). As expected, due to its stronger association with Ubc9, the RanGAP1$^{EE}$ mutant protein displayed an increased ratio of RanGAP1-SUMO1 to RanGAP1 relative to the two other RanGAP1 proteins (*Figure 5E*). Similar result was also observed when HA-RanGAP1$^{EE}$ transfected cell lysates was blotted with RanGAP1 antibody instead of HA antibody (*Figure 5—figure supplement 1A*). These results reveal that PKC-θ-mediated phosphorylation of RanGAP1 is essential for assembly of the RanBP2/RanGAP1-SUMO1/Ubc9 subcomplex in NPCs. Intriguingly, in silico analysis showed that replacement of the two PKC-θ phosphorylation sites in RanGAP1 by glutamic acid (S504E/S506E) increased the overall structural stability of RanGAP1 (*Figure 5—figure supplement 1B*); this increased stability likely contributed to the observed increased association between RanGAP1 and Ubc9. Collectively, these findings demonstrate that PKC-θ phosphorylates RanGAP1 to increase its association with Ubc9 and, in turn, the sumoylation of RanGAP1, thereby facilitating assembly of the RanBP2/RanGAP1-SUMO1/Ubc9 subcomplex upon TCR stimulation.

## RanGAP1$^{AA}$ mutant inhibits TCR/CD28-induced AP-1, NF-ATc1, and NF-κB nuclear import and IL-2 production

Based on the results above (*Figures 4* and *5*), we hypothesized that the non-phosphorylatable RanGAP1 mutant (RanGAP1$^{AA}$) will have impaired ability to promote nuclear transport of key TCR-activated transcription factors that are required for productive T-cell activation. As a positive control, we generated a RanGAP1 knockdown Jurkat E6.1 cell line (RanGAP1-KD) having a RanGAP1 mutation with decreased RanGAP1 protein level due to an in-frame nucleotide deletion (*Figure 6—figure*

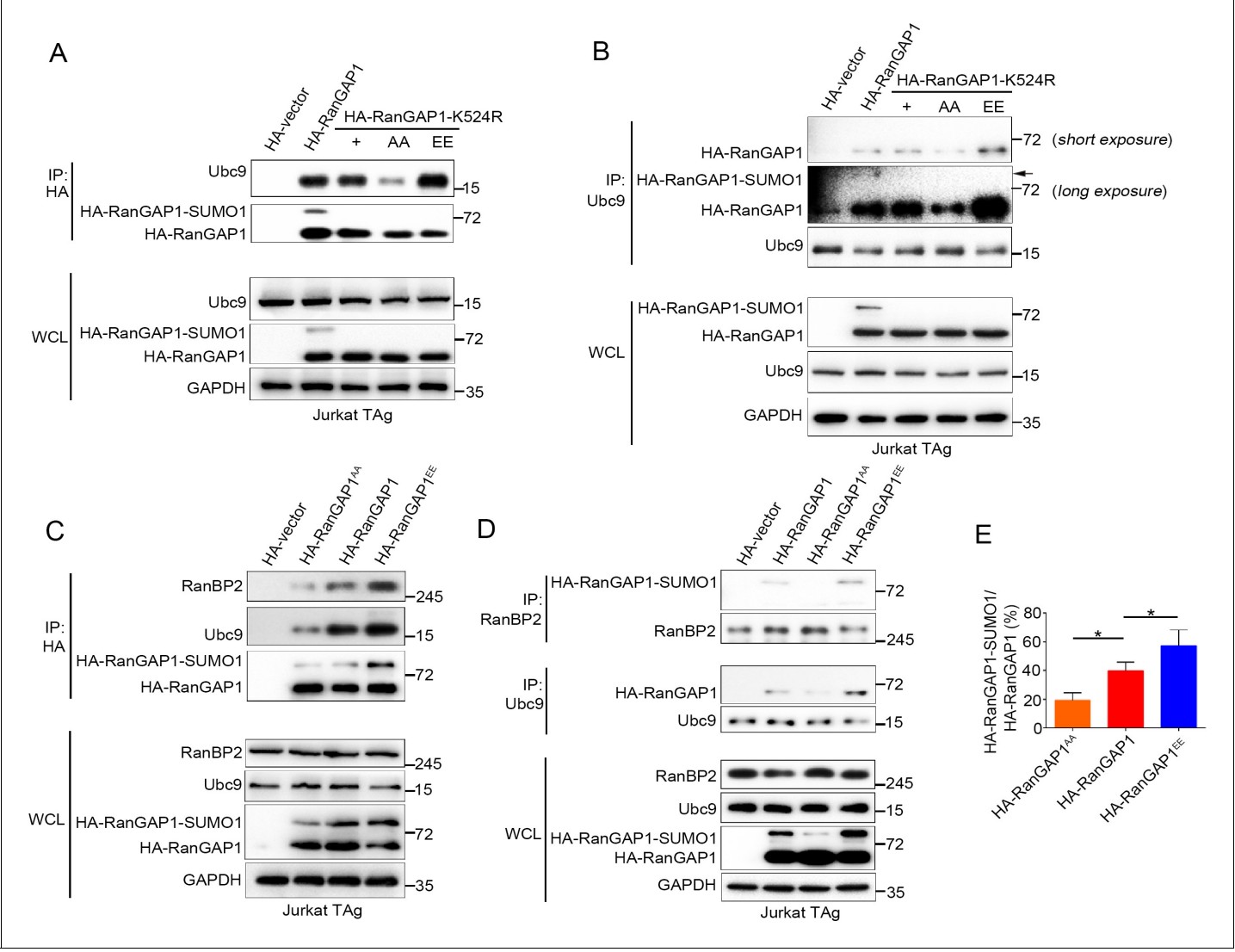

**Figure 5.** PKC-θ-mediated phosphorylation of RanGAP1 is required for its association with Ubc9 and RanBP2. (A, B) Reciprocal IP analysis of the association between HA-tagged wild-type or K524-mutated RanGAP1 and endogenous Ubc9. RanGAP1 expression was analyzed by anti-HA antibody immunoblotting in WCL (bottom panels). RanGAP1$^{AA}$ (S504A/S506A); RanGAP1$^{EE}$ (S504E/S506E). (C–E) Immunoblot analysis of HA-RanGAP1 IPs (C) and Ubc9 IPs or RanBP2 IPs (D) from Jurkat-TAg cells transfected with HA-RanGAP1 and HA-RanGAP1 mutants. The ratio of RanGAP1-SUMO1 to RanGAP1 in the WCL of (C) and (D) is quantified in (E); quantification is based on three biological replicates. *p<0.05 (one-way ANOVA with post hoc test). Data are representative of three biological replicates.

The online version of this article includes the following source data and figure supplement(s) for figure 5:

**Source data 1.** Uncropped western blot for *Figure 5*.

**Source data 2.** Row data for *Figure 5*.

**Figure supplement 1.** Characterization of the effect of Ser/Glu mutation on RanGAP1.

**Figure supplement 1—source data 1.** Uncropped western blot for *Figure 5—figure supplement 1*.

*supplement 1A,B*). We confirmed that expression of this mutant resulted in blocked TCR-induced nuclear translocation of NFATc1, p65 (NF-κB), and c-Jun and c-Fos (AP-1) (*Figure 6—figure supplement 1C,D*).

We next determined whether transfection of the RanGAP1-KD cell line with wild-type RanGAP1 or RanGAP1$^{AA}$ can rescue the nuclear translocation of NFATc1, p65, or AP-1, with unedited Jurkat cells serving as a negative control. Using subcellular fractionation (*Figure 6A*) or confocal microscopy (*Figure 6B,C*), we observed that the defective nuclear translocation of these transcription factors in RanGAP1-KD cells was largely rescued by wild-type RanGAP1, but not by the RanGAP1$^{AA}$ mutant.

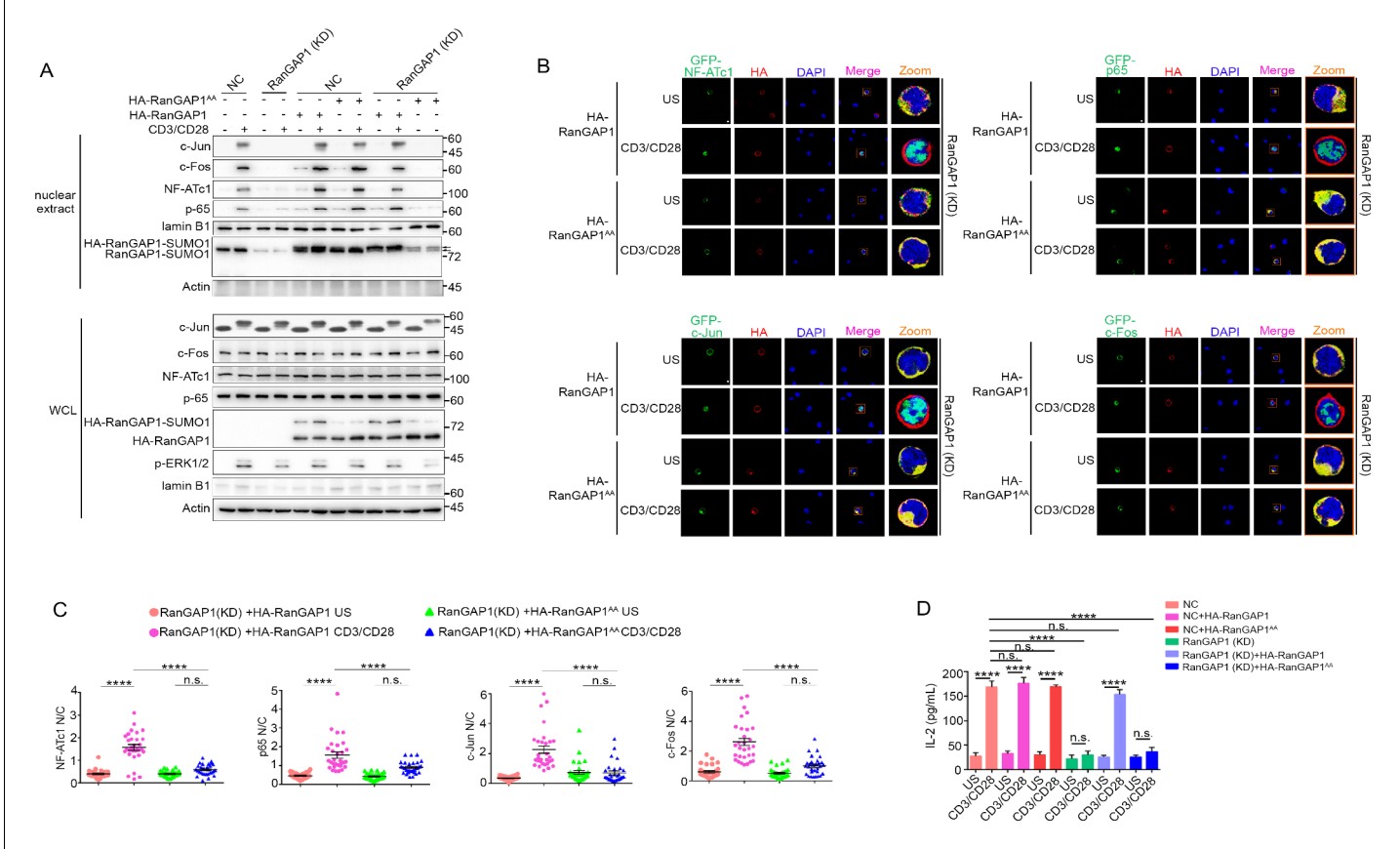

**Figure 6.** Wild-type RanGAP1, but not RanGAP1[AA], rescues deficient TCR-induced nuclear import of NF-ATc1, NF-κB, AP-1 in RanGAP1 (KD) Jurkat T cells. (**A**) Immunoblot analysis of NC (transfected with CRISPR/Cas9 vector expressing a scrambled gRNA) and RanGAP1(KD) (containing an in-frame nucleotide deletion in RanGAP1 gene induced by CRISPR/Cas9 genome editing) Jurkat E6.1 cells transfected with HA-RanGAP1 or HA-RanGAP1[AA], which were left unstimulated or stimulated for 15 min with anti-CD3 plus anti-CD28, followed by nuclear fractionation and immunodetection with the indicated antibodies. RanGAP1[AA] (S504A/S506A). (**B**) Confocal imaging of the nuclear import of GFP-tagged NF-ATc1, p65 (NF-κB), c-Jun and c-Fos in RanGAP1 (KD) cells cotransfected with HA-RanGAP1 or HA-RanGAP1[AA], and left unstimulated or stimulated with anti-CD3 plus anti-CD28. Scale bars, 3 μm. (**C**) Quantification of the nuclear import of GFP-tagged NF-ATc1, p65 (NF-κB), c-Jun and c-Fos in ~30 cells from two biological replicates as presented in (**B**). Each symbol represents an individual T cell. Horizontal lines indicate the mean ± s.e.m. n.s., not significant, ****p<0.0001 (one-way ANOVA with post hoc test). (**D**) Enzyme-linked immunosorbent assay of IL-2 in supernatants of NC or RanGAP1 (KD) cells transfected with HA-RanGAP1 or HA-RanGAP1[AA] and left unstimulated or stimulated for 24 hr with anti-CD3 plus anti-CD28. n.s., not significant, ****p<0.0001 (one-way ANOVA with post hoc test). Data are representative of three (**A, D**) or two (**B, C**) biological replicates.

The online version of this article includes the following source data and figure supplement(s) for figure 6:

**Source data 1.** Uncropped western blot for *Figure 6*.

**Source data 2.** Row data for *Figure 6* and for *Figure 6—figure supplement 1*.

**Figure supplement 1.** RanGAP1 is required for TCR-induced nuclear import of NF-ATc1, NF-κB, and AP-1.

**Figure supplement 1—source data 1.** Uncropped western blot for *Figure 6—figure supplement 1*.

Similarly, while stimulated RanGAP1-KD cells displayed reduced IL-2 production, which is known to require the cooperative activity of the above transcription factors, as compared to control cells, transfection with wild-type RanGAP1 rescued IL-2 production, while RanGAP1[AA] did not (*Figure 6D*). Moreover, by knocking down endogenous RanGAP1 with two different siRNAs designed to specifically target the 3'-UTR of RanGAP1 (*Figure 6—figure supplement 1E*), we validated the effect of the RanGAP1-AA mutation on TCR-induced nuclear translocation of the three transcription factors in human primary T cells (*Figure 6—figure supplement 1E-G*).

## PKC-θ-RanGAP signaling axis is differentially required in various nuclear transport pathways

To determine whether the nuclear transport defect resulting from PKC-θ deficiency is a generalized effect, we compared the nuclear transport of additional proteins in wild-type or PKC-θ knockdown Jurkat T cells (*Figure 7*). The proteins we tested were as follows: RNA exosome complex subunit Dis3, ribosome subunits RPS3 and RPL26, tumor suppressor p53, histone H1, which are importin-β1-dependent; histones (H2B, H3), protein/RNA export receptor CRM1, which are Ran-dependent but importin-β1-independent; and Ran-independent mRNA export receptor NXF1 (*Bernardes and Chook, 2020*; *Serpeloni et al., 2011*).

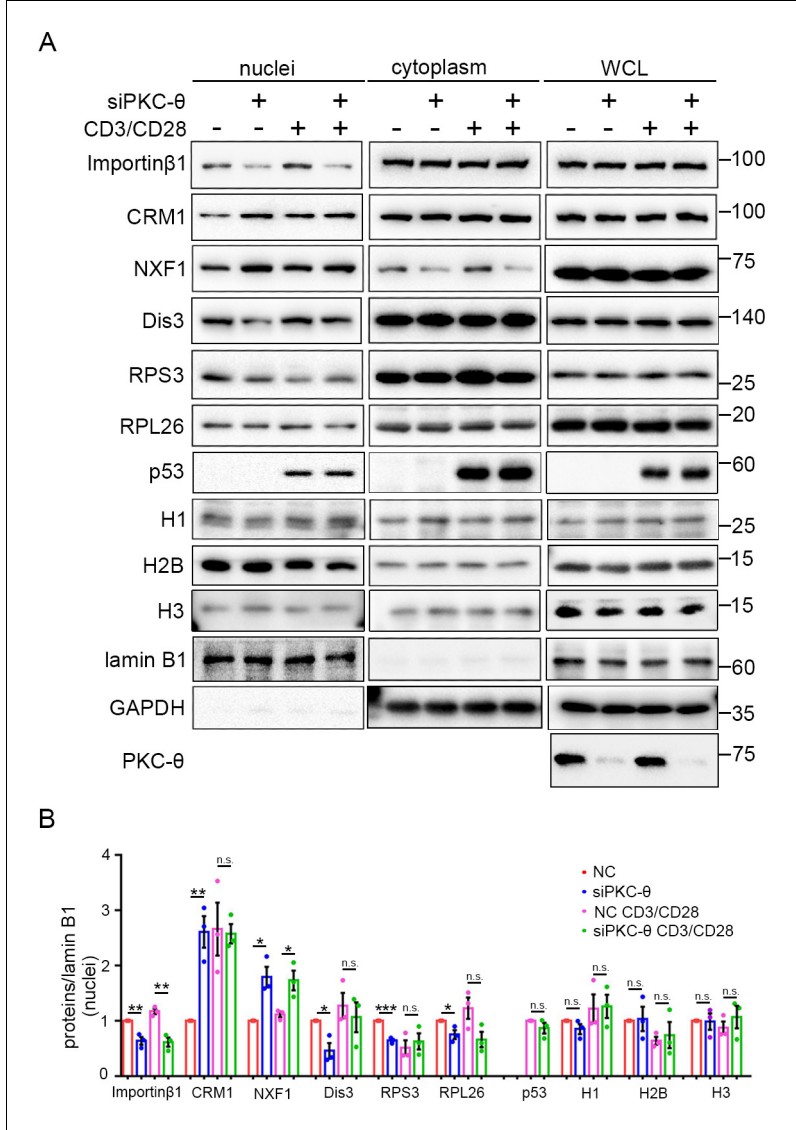

**Figure 7.** PKC-θ-RanGAP signaling axis is differentially required in various nuclear transport pathways. (**A**) Subcellular fractionation of Jurkat E6.1 cells transfected with siNC or siPKC-θ and stimulated for 0–15 min with anti-CD3 plus anti-CD28, followed by immunodetection with the indicated antibodies. (**B**) Statistical analysis of the indicated proteins in nuclear extract from the experiment of (**A**). n.s., not significant, *p<0.05, **p<0.01, ***p<0.001 (two-tailed, unpaired Student's t-test). Data are representative of three biological replicates. The online version of this article includes the following source data for figure 7:

**Source data 1.** Uncropped western blot for *Figure 7*.
**Source data 2.** Row data for *Figure 7*.

In resting T cells, PKC-θ knockdown led to reduced nuclear import of Dis3, RPS3, and RPL26, as well as impaired nuclear export of CRM1, consistent with their requirement for importin-β1 or the Ran system; however, the Ran-independent NXF1 also displayed impaired nuclear export (i.e., nuclear retention) (*Figure 7A,B*). In TCR-activated cells, PKC-θ knockdown did not affect the nuclear transport of CRM1, Dis3, RPS3, or RPL26 (*Figure 7A,B*). However, TCR stimulation did not alleviate siPKC-θ-caused defects in importin-β1 nuclear import and in NXF1 nuclear export. Interestingly, independent of the stimulation status, PKC-θ knockdown did not affect the nuclear transport of histones, nor did it affect the nuclear import of stimulation-induced p53. In summary, neither PKC-θ nor TCR signaling regulates histones transport; in resting T cells, the importin-β1- or Ran-dependent nuclear transport pathways we tested, with the exception of histones, are dependent on PKC-θ; in activated cells, PKC-θ was dispensable for their nuclear transport, except for the nuclear transport of importin-β1; and PKC-θ was required for NXF1 transport under both statuses. Therefore, PKC-θ-Ran-GAP signaling axis is differentially required in various nuclear transport pathways.

## RanGAP1$^{EE}$ mutant rescues c-Jun nuclear import in PKC-$\theta$-deficient T cells upon TCR stimulation

To further confirm our finding that the defective nuclear import of AP-1, NF-AT, and NF-κB transcription factors in PKC-θ-deficient cells is linked to the impaired phosphorylation of RanGAP1, we next determined whether a phospho-mimic RanGAP1$^{EE}$ mutant can rescue the nuclear translocation of c-Jun (AP-1), NF-ATc1 (NFAT), or p65 (NF-κB) in PKC-θ-deficient T cells. We transfected PKC-θ knockdown Jurkat T cells or retrovirally transduced murine splenic *Prkcq*$^{-/-}$ T cells with RanGAP1 EE or the negative control AA mutant expression vectors and analyzed their effect. As expected, the nuclear import of importin-β1 was partially rescued by RanGAP1$^{EE}$ both before or after TCR stimulation in siPKC-θ T cells (*Figure 8A*), consistent with our results that PKC-θ is required for RanGAP1 SUMOylation in both states (*Figures 1G–L* and *3D–F*). Moreover, in contrast to RanGAP1$^{AA}$ mutant, RanGAP1$^{EE}$ mutant did rescue TCR-induced nuclear translocation of c-Jun in siPKC-θ Jurkat T cells (*Figure 8A*) or in *Prkcq*$^{-/-}$ primary T cells (*Figure 8B,C*). The incomplete rescue of c-Jun translocation may be explained by other NPC functional defects caused by PKC-θ deficiency. As expected, NFATc1 or p65 nuclear import was not rescued by the RanGAP1$^{EE}$ mutant, consistent with the fact that TCR-proximal upstream signaling pathways are also disrupted in PKC-θ-deficient T cells (*Pfeifhofer et al., 2003*; *Sun et al., 2000*). As to the mRNA export receptor NXF1, siPKC-θ-induced nuclear retention (*Figure 7*) was reversed by forced RanGAP1$^{EE}$ expression (*Figure 8A*). Thus, we conclude that formation of the RanBP2 subcomplex is the key step downstream of PKC-θ signaling that regulates the nuclear transport of transcription factors, especially c-Jun.

## Discussion

PKC-θ plays an indispensable role in T-cell activation, including the TCR-induced activation of NF-κB, AP-1, and NFAT, the main transcription factors required for acquisition of effector functions and cytokine production of T cells (*Altman and Kong, 2016*; *Pfeifhofer et al., 2003*; *Sun et al., 2000*). These PKC-θ-dependent functions depend on its recruitment to the T-cell IS and its association with CD28 (*Kong et al., 2011*). However, little is known about potential nuclear functions of PKC-θ, with the exception of a few studies that demonstrated a role for nuclear and chromatin-associated PKC-θ in promoting gene expression (*Li et al., 2016*; *Sutcliffe et al., 2011*). Here, we have demonstrated that upon TCR plus CD28 costimulation, PKC-θ phosphorylates RanGAP1 to promote its interaction with Ubc9 and increase the sumoylation of RanGAP1, which, in turn, enhances assembly of the RanBP2 subcomplex and, thus, directly promotes the nuclear import of AP-1, NFAT, and NF-κB (*Figure 9*). Thus, our work reveals a novel signaling axis, TCR-PKC-θ-RanGAP1, which regulates T-cell activation via control of nucleo-cytoplasmic transport, thereby linking TCR signaling to formation of the NPC.

Evidence for crosstalk between NPC components and immunological signaling is emerging (*Borlido et al., 2018*; *Gu et al., 2016*; *Liu et al., 2015*), yet much remains unknown. While the sumoylation of RanGAP1 is essential for the assembly and function of the RanBP2 complex (*Matunis et al., 1998*; *Saitoh et al., 1997*), the signals that regulate this sumoylation are unclear. The sumoylation of target proteins is generally highly transient, but RanGAP1 is an exception in that it is constitutively sumoylated (*Mahajan et al., 1997*; *Saitoh et al., 1997*), a result of Ubc9 directly

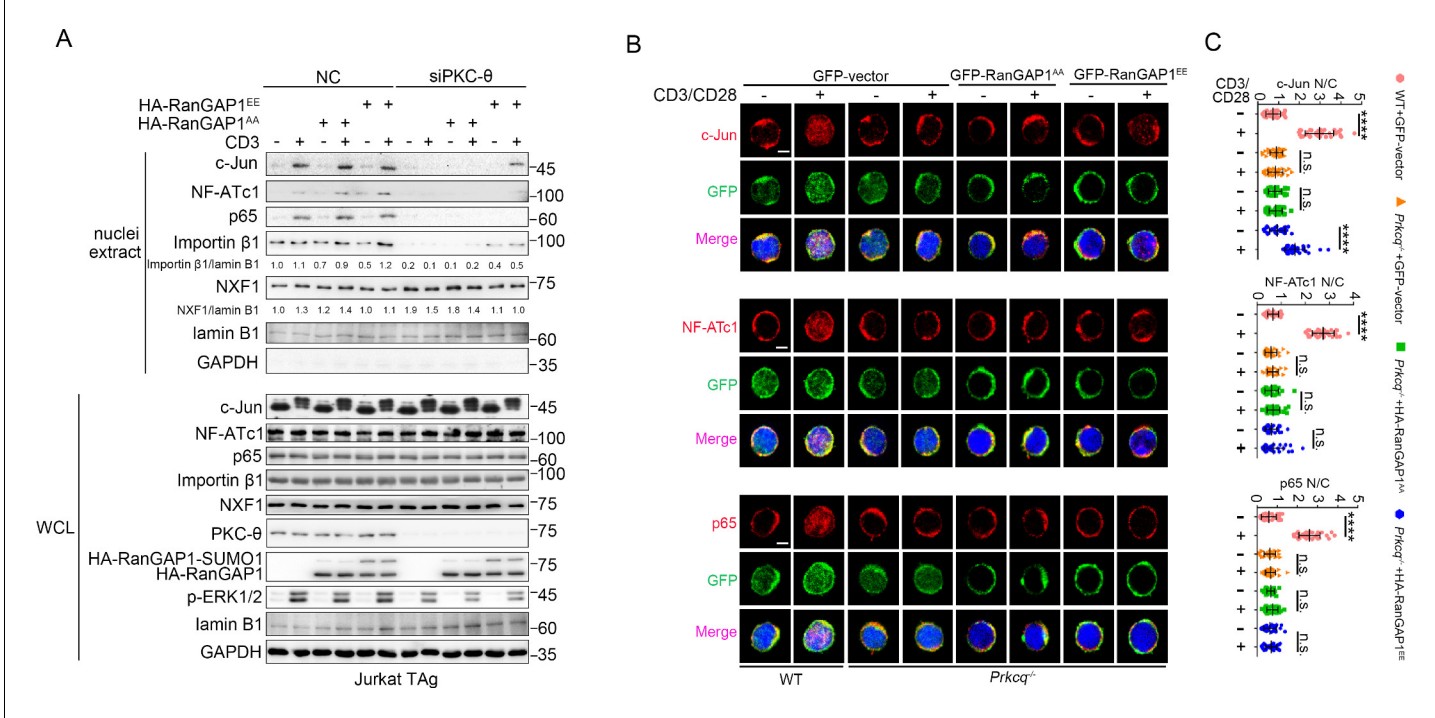

**Figure 8.** RanGAP1[EE], but not RanGAP1[AA], rescues TCR-induced nuclear import of c-Jun in PKC-θ-deficient cells. (**A**) Immunoblot analysis of Jurkat-TAg cells cotransfected with siNC or siPKC-θ plus HA-RanGAP1[AA] or HA-RanGAP1[EE] as indicated, which were left unstimulated or stimulated for 15 min with anti-CD3, followed by nuclear fractionation and immunodetection with the indicated antibodies. (**B**) Confocal imaging showing the nuclear import of c-Jun, NF-ATc1 or p65 (NF-κB) in wild-type (WT) murine splenic T cells transduced with GFP-vector or *Prkcq*[−/−] murine splenic T cells transduced with GFP-vector or RanGAP1[AA] or RanGAP1[EE], left unstimulated or stimulated with anti-CD3 plus anti-CD28 for 15 min. Scale bars, 2 μm. These images are representative of ~30 cells analyzed in each group in two independent experiments. (**C**) Quantification of the nuclear import of c-Jun, NF-ATc1, or p65 as presented in (**B**). Each symbol represents an individual T cell. Horizontal lines indicate the mean ± s.e.m. n.s., not significant, ****p<0.0001 (one-way ANOVA with post hoc test). Data are representative of two biological replicates.

The online version of this article includes the following source data for figure 8:

**Source data 1.** Uncropped western blot for *Figure 8*.

**Source data 2.** Row data for *Figure 8*.

recognizing and catalyzing sumoylation of the RanGAP1 φ-K-x-D/E consensus motif at amino acid residues 525–528 (*Bernier-Villamor et al., 2002*; *Lee et al., 1998*; *Zhu et al., 2006*). The inter-regulation between phosphorylation and sumoylation is a common mechanism for the regulation of protein function (*Hietakangas et al., 2006*; *Tomasi and Ramani, 2018*). Several Ser/Thr phosphorylation sites have been identified in RanGAP1, including the phosphorylation of Ser[358] by casein kinase II kinase to promote RanGAP1 binding to Ran protein (*Takeda et al., 2005*), phosphorylation of Thr[409], which is related to the nuclear accumulation of cyclin B1 (*Swaminathan et al., 2004*), and mitotic phosphorylation of Ser[428] and Ser[442] with unknown function (*Swaminathan et al., 2004*). However, none of these phosphorylations affect the sumoylation of RanGAP1. In contrast, phosphorylation of Ser[504] and Ser[506] by PKC-θ, which we documented here, promoted the sumoylation of RanGAP1 by enhancing the interaction between RanGAP1 and Ubc9, which may be facilitated by these two phosphorylation sites being close to the φ-K-x-D/E motif. Thus, our study suggests a unique role for PKC-θ in linking TCR signaling to assembly of the RanBP2 complex.

Our finding that PKC-θ deficiency essentially abolished the binding of RanGAP1-SUMO1 to NPC while only moderately inhibiting the sumoylation of RanGAP1 upon stimulation in vivo (*Figure 3D,E*) is of interest. A possible explanation for this apparent discrepancy is that the increased RanGAP1 phosphorylation itself, mediated by PKC-θ upon stimulation, may also assist the binding. Moreover, other PKC-θ substrates in addition to RanGAP1 may exist in the NPC complex; thus, phosphorylation of these proteins by PKC-θ may contribute to the TCR-induced NPC assembly as well. The T-cell

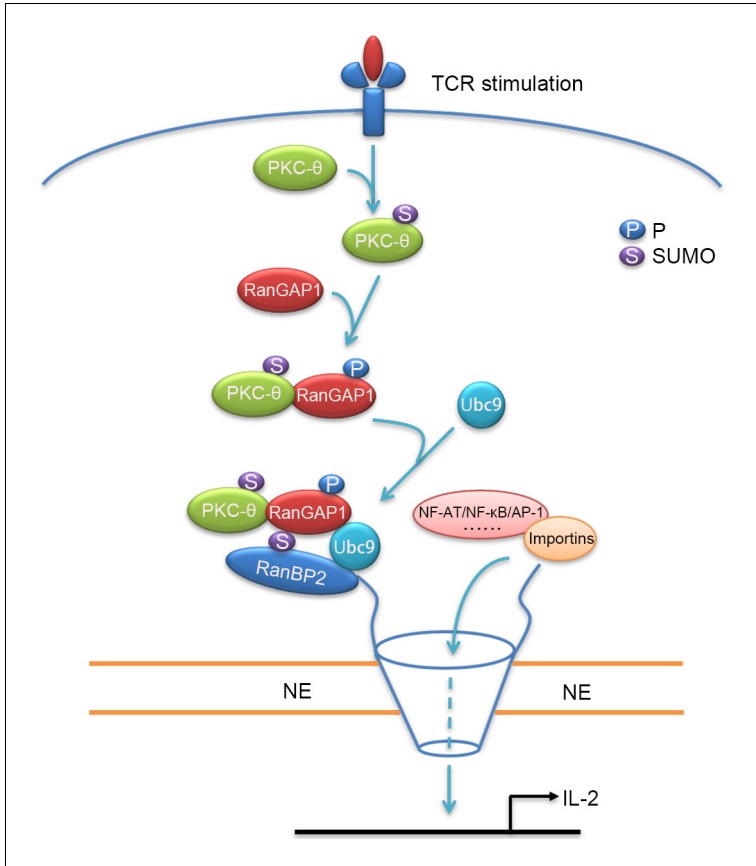

**Figure 9.** Schematic model of nuclear import regulation by the TCR-PKC-θ-RanGAP1 axis. Upon TCR stimulation, PKC-θ phosphorylates RanGAP1 to increase its association with Ubc9, thereby enhancing the sumoylation of RanGAP1, which is required for assembly of the RanBP2/RanGAP1-SUMO1/Ubc9 subcomplex. This complex then promotes the nuclear import of NF-ATc1, NF-κB, and AP-1.

adaptor protein, SLP76, has been reported to bind to RanGAP1-SUMO1 and to promote the TCR-induced nuclear import of NFAT (*Liu et al., 2015*). Combining these findings, we propose that some TCR signaling modules, e.g., PKC-θ and SLP76, translocate to the NE to regulate nucleo-cytoplasmic transport and, furthermore, that PKC-θ plays an indispensable role in NPC assembly.

One explanation for our observation that generally importin-β1 or Ran-dependent nuclear transport of proteins was mostly unaffected in PKC-θ-deficient cells (*Figure 7*) is that other transport pathways may play a compensatory role in the absence of PKC-θ, especially under stimulation conditions. The RanBP2/RanGAP1-SUMO1/Ubc9 subcomplex is thought to have evolved to facilitate efficient nuclear transport but not a prerequisite complex for nuclear transport per se because RanBP2 is only found in the animal kingdom and SUMO modification does not apply to plant RanGAP1 (*Hutten et al., 2008*). In addition, soluble RanGAP1 may function as a substitute for RanBP2-associated RanGAP1-SUMO1 to promote nuclear transport (*Hutten et al., 2008*). Of note, in both resting and activated siPKC-θ T cells, RanGAP1-SUMO1 was indispensable for the nuclear import of importin-β1, indicating that the integrity of the RanBP2 subcomplex is essential for importin-β1 nuclear import, consistent with previous observations (*Hutten et al., 2008*). Interestingly, the Ran-independent mRNA export receptor NXF1 was also retained in the nucleus in PKC-θ knockdown cells and this retention could be reversed by forced RanGAP1[EE] expression, suggesting that a some other activity (the SUMO E3 ligase activity) other than the GTPase activity of RanBP2 subcomplex is involved (*Ritterhoff et al., 2016*).

Despite the fact that PKC-θ is required for TCR-induced T-cell activation in vitro (*Pfeifhofer et al., 2003*; *Sun et al., 2000*), PKC-θ deficiency has a different impact on immune responses against in vivo bacterial (*Sakowicz-Burkiewicz et al., 2008*), viral (*Berg-Brown et al.,*

*2004*; *Giannoni et al., 2005*; *Marsland et al., 2005*; *Marsland et al., 2004*) and helminth infections or model allergens (*Marsland et al., 2004*; *Salek-Ardakani et al., 2005*). Specifically, PKC-θ is required for Th2 cell differentiation and Th2-mediated immune responses against helminth and for the proliferation and survival of pathogen-specific T Cells in murine listeriosis (*Marsland et al., 2004*; *Sakowicz-Burkiewicz et al., 2008*). By contrast, PKC-θ is largely dispensable for Th1 differentiation and CD4 Th1/CD8 T cells-mediated resistance against viral and intracellular protozoan *Leishmania major* infection (*Berg-Brown et al., 2004*; *Giannoni et al., 2005*; *Marsland et al., 2005*; *Marsland et al., 2004*). One possible explanation is that strong signals including high antigen dose, TLR ligands, and proinflammatory cytokines (such as IL-1 and TNFα) produced during Th1 and CD8 T-cell responses may well compensate for the loss of PKC-θ signaling (*Altman and Kong, 2016*; *Manicassamy et al., 2006*; *Marsland and Kopf, 2008*). Considering the phosphorylation defect of RanGAP1 in PKC-θ deficient T cells, it might be reversed by a compensatory kinase-mediated phosphorylation of RanGAP1 in response to strong signals. In addition, we speculated the cargo selectivity of importins in regulation of nuclear transport might also contribute to the different requirements for PKC-θ in various T-cell immune responses. It is known that STAT1 transcription factor mediates IFN-γ signaling for Th1 differentiation while AP-1 transcription factors are required for IL-4/10 production in Th2 differentiation (*Li et al., 1999*; *Wang et al., 2005*). It has also been demonstrated that the nuclear import of AP-1 requires importin-β and Ran but is independent of importin-α (*Forwood et al., 2001*),whereas STAT1 can constitutively transport into the nucleus independent of importin-β (*Meyer et al., 2002*). Thus, the PKC-θ-induced association of RanGAP1 with importin-β might be less important in IFN-γ signaling of Th1 cells but is indispensable in IL-4/10 signaling of Th2 cells. Further studies by rescuing the function of RanGAP1 in *Prkcq*$^{-/-}$ mice may help to understand the differential requirement for PKC-θ in T cells immune responses.

During thymocyte-positive selection, PKC-θ is not required for TCR-induced activation of NF-κB and calcium signaling, which may be compensated by other PKC isoforms, such as PKCη (*Altman and Kong, 2016*; *Fu et al., 2011*; *Morley et al., 2008*; *Sun et al., 2000*). Similarly, in contrast to mature T cells, TCR-induced assembly of RanBP2 subcomplex and nuclear import of c-Jun are relatively intact in *Prkcq*$^{-/-}$ thymocytes (*Figure 3D,F,J*), confirming the existence of compensation for PKC-θ in thymocytes. However, the irreplaceable role of PKC-θ in TCR-induced activation of ERK endows PKC-θ a critical molecule in the positive selection as well as in the activation of mature T cells (*Altman and Kong, 2016*; *Morley et al., 2008*). Consistently, we also observed the defect of TCR-induced ERK activation caused by PKC-θ deficiency cannot be compensated in thymocytes (*Figure 3F,J*). Therefore, PKC-θ is essential for TCR-induced ERK activation both in mature and immature T cells. Due to the important roles of ERK signal in both αβ T and γδ T-cell development (*Ciofani and Zúñiga-Pflücker, 2010*; *Delgado et al., 2000*; *Fischer et al., 2005*; *McNeil et al., 2005*), it will be interesting to further investigate the role of PKC-θ in γδ T cells development.

Upon TCR stimulation, PKC-θ also translocates to the nucleus, where it associates with chromatin and forms transcription complexes with Pol II, MSK-1, LSD1, and 14-3-3ζ (*Sutcliffe et al., 2011*) or phosphorylates a key splicing factor, SC35, and histone (*Li et al., 2016*; *McCuaig et al., 2015*) to activate downstream gene transcription. Here, we show that sumoylated PKC-θ translocates to the NE to promote the function of the NPC. Interestingly, while the nuclear localization sequence of PKC-θ is known to mediate its nuclear localization (*Sutcliffe et al., 2012*), desumoylation of PKC-θ or inactivation of its catalytic activity prevented its NE localization (*Figure 2F*). Thus, different mechanisms may be involved in PKC-θ translocation to the NE vs. the nucleus. It is, therefore, possible that localization of PKC-θ to the NE promotes the function of NPC, which, in turn, would enable the nuclear translocation of cargo PKC-θ. We have previously demonstrated that PKC-θ sumoylation, catalyzed by the SUMO E3 ligase PIASxβ, is required for its central IS localization (*Wang et al., 2015*). It remains to be determined whether PKC-θ translocation to the NE similarly requires or is dependent on another mechanism.

The RanBP2 subcomplex is a multifunctional component of NPC in the animal kingdom (*Hampoelz et al., 2019a*; *Lin and Hoelz, 2019*). In addition to its roles in nuclear transport and NPC assembly (*Hutten et al., 2008*), the RanBP2 subcomplex has other cellular functions, including translational control (*Mahadevan et al., 2013*) and nuclear import of pathogens (*Dharan et al., 2016*). Moreover, the RanBP2/RanGAP1-SUMO1/Ubc9 complex is a multi-subunit SUMO E3 ligase (*Reverter and Lima, 2005*; *Werner et al., 2012*) that mediates the sumoylation of the GTPase Ran (*Hutten et al., 2008*; *Sakin et al., 2015*; *Wälde et al., 2012*) and the sumoylation of topoisomerase

IIa and borealin to regulate mitosis (*Dawlaty et al., 2008*; *Klein et al., 2009*). Therefore, our current findings that reveal a TCR-PKC-θ-RanGAP1 signaling axis not only exposes a novel regulatory layer of TCR signaling, but also provides a new angle to understand fundamental mechanisms of T-cell immunity.

# Materials and methods

## Key resources table

| Reagent type (species) or resource | Designation | Source or reference | Identifiers | Additional information |
|---|---|---|---|---|
| Strain, strain background (*Mus musculus*) | C57BL/6 (*Prkcq⁻ᐟ⁻*) | A gift from D. Littman (*Wang et al., 2015*) | PMID:26390157 | |
| Cell line (*Homo-sapiens*) | Jurkat, Clone E6-1 | ATCC | TIB-152 | |
| Cell line (*Homo-sapiens*) | Jurkat-TAg | Cellosaurus | CVCL_C831 RRID:CVCL_C831 | |
| Antibody | Goat polyclonal anti-PKC-θ | Santa Cruz Biotechnology | Cat #: sc-1875, RRID:AB_675806 | IF (1:200) |
| Antibody | Mouse monoclonal anti-RanGAP1 | Santa Cruz Biotechnology | Cat #: sc-28322, RRID:AB_2176987 | WB (1:1000) |
| Antibody | Mouse monoclonal anti-importin β1 | Santa Cruz Biotechnology | Cat #: sc-137016, RRID:AB_2133993 | WB (1:1000) IF (1:200) |
| Antibody | Mouse monoclonal anti- Ran | Santa Cruz Biotechnology | Cat #: sc-271376, RRID:AB_10610890 | WB (1:1000) IF (1:200) |
| Antibody | Mouse monoclonal anti- RanBP2 | Santa Cruz Biotechnology | Cat #: sc-74518, RRID:AB_2176784 | WB (1:1000) |
| Antibody | Mouse monoclonal anti- Ubc9 | Santa Cruz Biotechnology | Cat #: sc-271057, RRID:AB_10610674 | WB (1:1000) |
| Antibody | Mouse monoclonal anti- NF-ATc1 | Santa Cruz Biotechnology | Cat #: sc-7294, RRID:AB_2152503 | WB (1:1000) |
| Antibody | Mouse monoclonal anti-c-Myc | Santa Cruz Biotechnology | Cat #: sc-40, RRID:AB_2857941 | WB (1:1000) IF (1:200) |
| Antibody | Mouse monoclonal anti- actin | Santa Cruz Biotechnology | Cat #: sc-8432, RRID:AB_626630 | WB (1:1000) |
| Antibody | Mouse monoclonal anti-p-Ser/ Phosphoserine | Santa Cruz Biotechnology | Cat #: sc-81516, RRID:AB_1128626 | WB (1:1000) |
| Antibody | Goat polyclonal anti-Lamin B1 | Santa Cruz Biotechnology | Cat #: sc-30264, RRID:AB_2136305 | WB (1:1000) IF (1:200) |
| Antibody | Rabbit monoclonal anti-p65(NF-κB) | Santa Cruz Biotechnology | Cat #: sc-109, RRID:AB_632039 | WB (1:1000) IF (1:200) |
| Antibody | Mouse monoclonal anti-Histone 1 | Santa Cruz Biotechnology | Cat #: sc-8030, RRID:AB_675641 | WB (1:500) |
| Antibody | Rabbit monoclonal anti-c-Jun | Cell Signaling Technology | Cat #: 9165, RRID:AB_2130165 | WB (1:1000) IF (1:200) |
| Antibody | Mouse monoclonal anti-Dis3 | Santa Cruz Biotechnology | Cat #: sc-398663 | WB (1:1000) |

*Continued on next page*

*Continued*

| Reagent type (species) or resource | Designation | Source or reference | Identifiers | Additional information |
|---|---|---|---|---|
| Antibody | Rabbit monoclonal anti-c-Fos | Cell Signaling Technology | Cat #: 2250, RRID:AB_2247211 | WB (1:1000) IF (1:200) |
| Antibody | Rabbit monoclonal anti-phospho-Ser/Thr | Cell Signaling Technology | Cat #: 9631, RRID:AB_330308 | WB (1:1000) |
| Antibody | Mouse monoclonal anti-phospho-ERK1/2 | Cell Signaling Technology | Cat #: 9106, RRID:AB_331768 | WB (1:1000) |
| Antibody | Rabbit monoclonal anti-Na/K-ATPas | Cell Signaling Technology | Cat #: 3010, RRID:AB_2060983 | WB (1:1000) |
| Antibody | Rabbit monoclonal anti-GAPDH | Cell Signaling Technology | Cat #: 2118, RRID:AB_561053 | WB (1:1000) |
| Antibody | Rabbit monoclonal anti-CRM1 | Cell Signaling Technology | Cat #: 46249, RRID:AB_2799298 | WB (1:1000) |
| Antibody | Rabbit monoclonal anti-Histone 2B | Cell Signaling Technology | Cat #: 12364, RRID:AB_2714167 | WB (1:1000) |
| Antibody | Rabbit monoclonal anti-Histone 3 | Cell Signaling Technology | Cat #: 4499, RRID:AB_10544537 | WB (1:1000) |
| Antibody | Rabbit monoclonal anti-NF-ATc1 | Abcam | Cat #: ab25916, RRID:AB_448901 | IF (1:200) |
| Antibody | Rabbit monoclonal anti-RanGAP1 | Abcam | Cat #: ab92360, RRID:AB_10564003 | IF (1:200) |
| Antibody | Rabbit polyclonal anti-RPL26 | Abcam | Cat #: ab59567, RRID:AB_945306 | WB (1:2000) |
| Antibody | Rabbit monoclonal anti-RPS3 | Abcam | Cat #: ab181992 | WB (1:2000) |
| Antibody | Rabbit monoclonal anti-NXF1 | Abcam | Cat #: ab129160, RRID:AB_11142853 | WB (1:2000) |
| Antibody | Mouse monoclonal anti- Mab414 | BioLegend | Cat #: 902901, RRID:AB_2565026 | WB (1:1000) IF (1:200) |
| Antibody | Rat monoclonal anti-mouse CD3 | BioLegend | Cat #: 100202, RRID:AB_312659 | 5 µg/ml |
| Antibody | Syrian Hamster monoclonal anti-mouse CD28 | BioLegend | Cat #: 102102, RRID:AB_312867 | 2 µg/ml |
| Antibody | Mouse monoclonal anti-Human CD3(OKT3) | BioLegend | Cat #: 317302, RRID:AB_571927 | 5 µg/ml |
| Antibody | Mouse monoclonal anti-Human CD28(CD28.2) | BioLegend | Cat #: 302902, RRID:AB_314304 | 2 µg/ml |
| Antibody | Alexa Fluor 488-coupled chicken anti-mouse IgG | Invitrogen | Cat #: A-21200, RRID:AB_2535786 | IF (1:2000) |
| Antibody | Alexa Fluor 594-coupled chicken anti-mouse IgG | Invitrogen | Cat #: A-21201, RRID:AB_141630 | IF (1:2000) |
| Antibody | Alexa Fluor 594-coupled chicken anti-rabbit IgG | Invitrogen | Cat #: A-21442, RRID:AB_141840 | IF (1:2000) |

*Continued on next page*

*Continued*

| Reagent type (species) or resource | Designation | Source or reference | Identifiers | Additional information |
|---|---|---|---|---|
| Antibody | Alexa Fluor 488-coupled donkey anti-goat IgG | Invitrogen | Cat #: A-11055, RRID:AB_2534102 | IF (1:2000) |
| Other | Cell Tracker Blue | Invitrogen | Cat #: C2110 | IF: 10 μM |
| Recombinant DNA reagent | pcDNA3.1( ) (plasmid) | Invitrogen | Cat #: V79020 | |
| Recombinant DNA reagent | pGEX-4T-2 | GE | Cat #: 27-4581-01 | |
| Recombinant DNA reagent | pFlag-CMV2 | Sigma | Cat #: E7396 | |
| Recombinant DNA reagent | pMXs-IRES-GFP Retroviral Vector | Cell Biolabs | Cat #: RTV-013 | |
| Recombinant DNA reagent | LentiCRISPRv2 | Addgene | Cat #: 52961 | |
| Sequence-based reagent | RanGAP1-F | NM_001278651.2 | CGGGATCCATGGCCTCGGAAGACATTGCCAAGC | Primer for PCR |
| Sequence-based reagent | RanGAP1-R | NM_001278651.2 | ATAAGAATGCGGCCGCCTAGACCTTGTACAGCGTCTGCAGC | Primer for PCR |
| Sequence-based reagent | RanGAP1-S34A-F | NM_001278651.2 | CAAGAGCCTCAAACTCAACGCCGCAGAAGATGCTAAAGATG | Primer for PCR |
| Sequence-based reagent | RanGAP1-T419A-F | NM_001278651.2 | CTGGACCCTAACGCCGGGGAGCCAGCTC | Primer for PCR |
| Sequence-based reagent | RanGAP1-S478A-F | NM_001278651.2 | CCTTCCTAAAGGTGTCAGCCGTGTTCAAGGACGAAG | Primer for PCR |
| Sequence-based reagent | RanGAP1-S504A-F | NM_001278651.2 | GAAGGCTTTCAACGCCTCGTCCTTCAAC | Primer for PCR |
| Sequence-based reagent | RanGAP1-S506A-F | NM_001278651.2 | CTTTCAACTCCTCGGCCTTCAACTCCAAC | Primer for PCR |
| Sequence-based reagent | RanGAP1-S504A/S506A-F | NM_001278651.2 | CTGATGCAGAAGGCTTTCAACGCCAGCGCCTTCAACTCCAACACCTTCC | Primer for PCR |
| Sequence-based reagent | RanGAP1-S504E/S506E-F | NM_001278651.2 | CTGATGCAGAAGGCTTTCAACGAGAGCGAGTTCAACTCCAACACCTTCC | Primer for PCR |
| Sequence-based reagent | RanGAP1-K524R-F | NM_001278651.2 | CATGGGTCTGCTCAGGAGTGAAGACAAG | Primer for PCR |
| Sequence-based reagent | RanGAP1 sgRNA | NM_001278651.2 | CACCGCAGAGGGAGTGCCACT | CRISPR-Cas9 guides |
| Sequence-based reagent | shPKC-θ | NM_006257.5 | GAGTATGTCGAATCAGAGA | dsRNA for RNAi |
| Sequence-based reagent | siPKC-θ | Previous study in lab (*Wang et al., 2015*) | GCUUGUAACUUGAGAUCUA | dsRNA for RNAi |
| Sequence-based reagent | siRanGAP1-1 | NM_001278651.2 | GGAGUGUUGACAACCCAAA | dsRNA for RNAi |

*Continued on next page*

*Continued*

| Reagent type (species) or resource | Designation | Source or reference | Identifiers | Additional information |
|---|---|---|---|---|
| Sequence-based reagent | siRanGAP1-2 | NM_001278651.2 | GUGAGCUGCUC CGCCAUUAAA | dsRNA for RNAi |
| Software algorithm | Fiji/Image-J | MPI-CBG, Dresden/ National Institutes of Health (NIH) | PMID:22743772 RRID:SCR_002285 | Image processing and analysis |
| Software algorithm | Graphpad Prism v6 | Graphpad | RRID:SCR_002798 | Graphs and statistical analysis |

## Mice

C57BL/6 (B6) and *Prkcq*$^{-/-}$ mice (a gift from D. Littman) were housed under specific pathogen-free conditions and were manipulated according to the guidelines approved by the Animal Care and Ethics committee of Sun Yat-Sen University.

## Plasmids

The cDNAs encoding RanGAP1, p65/NF-κB, NF-ATc1, c-Jun, c-Fos were amplified by PCR from a Jurkat E6.1 T-cell cDNA library and were cloned into the vectors pcDNA3.1-HA (Invitrogen), pGEX-4T-2 (Sigma), or pcDNA3.1-GFP (Invitrogen). Plasmids encoding HA-tagged SUMO1, and c-Myc-tagged PKC-θ, PKC-θ−2KR, PKC-θ-K409R, PKC-θ-A148E have been described (*Wang et al., 2015*). Specific point mutations of RanGAP1 were introduced by site-directed mutagenesis with a Quik-Change Site-Directed Mutagenesis Kit (Stratagene).

## Human primary T cells

Buffy coat cells from de-identified healthy human peripheral blood was provided by the Guangzhou blood center; it was handled according to the guidelines of the Ethics committee of Sun Yat-Sen University. Peripheral blood mononuclear cell isolation and T-cell enrichment were performed as previously described (*Wang et al., 2015*).

## Cell culture, transfection, and stimulation

Spleens and thymi of *Prkcq*$^{-/-}$ mice were dissociated into single-cell suspensions in PBS containing 1% FBS (Gibco), and samples were depleted of red blood cells with RBC lysis buffer (Sigma). Mouse splenic T cells were isolated with a Pan T Cell Isolation Kit II (Miltenyi Biotec). Jurkat T cells and Raji B cells were cultured in complete RPMI-1640 (Hyclone, Logan, UT) medium supplemented with 10% FBS, and 100 U/ml each of penicillin and streptomycin (Life Technologies). Jurkat E6.1 T cells stably expressing PKC-θ-specific short hairpin RNA (shPKC-θ) were grown in the presence of aminoglycoside G418 (700 µg/ml; Invitrogen). Cell lines were electro-transfected with various vectors using the Cell Line Nucleofector Kit (Lonza, Germany), and human primary T cells were transfected with various vectors using the P3 Primary Cell 4D-Nucleofector Kit (Lonza). For APC stimulation of T cells, Raji B lymphoma cells were incubated for 30 min at 37°C in the presence or absence of SEE (100 ng/ml; Toxin Technology). The cells were washed with PBS and were mixed with Jurkat T cells at a ratio of 1:1, followed by incubation for various times at 37°C. For antibody stimulation, mouse or human T cells were stimulated for various times with anti-CD3 (5 µg/ml) and/or anti-CD28 (2 µg/ml), which were crosslinked with goat anti-mouse IgG (10 µg/ml). Cell lines were authenticated using short tandem repeat profiling at GENEWIZ, Inc (Suzhou, China). Mycoplasma test for cell culture was done in a monthly basis using PCR Mycoplasma Detection Kit (abm, G238). Cells used in experiments were within 10 passages from thawing.

## IP and immunoblotting

Cells washed with ice-cold PBS and lysed in lysis buffer (20 mM Tris–HCl, pH 7.5, 150 mM NaCl, 5 mM EDTA, 1% Nonidet-P-40, 5 mM NaPPi, 1 mM sodium orthovanadate [Na$_3$VO$_4$], 1 mM PMSF, and 10 µg/ml each aprotinin and leupeptin). Whole-cell lysates were incubated overnight at 4°C with

the indicated antibodies, and proteins were collected on protein G-Sepharose beads (GE Healthcare) for an additional 4 hr at 4°C with gentle shaking. The immunoprecipitated proteins were resolved by SDS–PAGE, transferred onto PVDF membranes, and probed with primary antibodies. Signals were visualized by enhanced chemiluminescence (YESEN, Shanghai, CHINA) and films were exposed in the ChemiDoc XRS system (Bio-Rad) or to X-ray film. Densitometry analysis was performed with ImageJ software.

## PM, NE, and NPC fractionation

For the PM purification (*Figure 1A,B*), cells washed with ice-cold PBS were resuspended in an ice-cold hypotonic buffer (10 mM HEPES, pH 7.4, 42 mM KCl, 5 mM MgCl$_2$, 1 mM DTT, and protease inhibitors [leupeptin, PMSF, and aprotinin]) for 20 min on ice. The cell suspension was pushed 10 times through a 30-gauge needle, followed by centrifugation at 200 × g for 10 min. The pellet as was used as the nuclear fraction. The supernatant was centrifugation at 25,000 × g for 1 hr, and the pellet was used as the PM.

The nuclear and NE fractions were prepared using the NE-PERTM Nuclear and Cytoplasmic Extraction kit (Thermo Scientific) as per manufacturer's instructions. Briefly, cells washed with ice-cold PBS were resuspended in Cytoplasmic Extraction Reagents containing proteinase inhibitors, vortexed vigorously, and centrifuged at 16,000 × g for 10 min. The pellet was resuspended in Nuclear Extraction Reagent containing proteinase inhibitors, vortexed vigorously, and centrifuged at 16,000 × g for 10 additional min. The supernatant and the insoluble fraction, representing the nuclear extract and NE, respectively, were collected.

The NPC fraction was prepared as described (*Jafferali et al., 2014*), with minor modifications. Cells were washed with PBS and treated with 1 mM dithiobis (succinimidyl propionate) (Sangon Biotech) in RPMI-1640 medium for 15 min at room temperature to crosslink the NPC. The reaction was stopped by adding 15 mM Tris–HCl (pH 7.4) for 10 min at room temperature. Nuclei were pelleted as described above, followed by incubation in 5 volumes of 7 M urea containing 1% Triton X-100 (TX-100) and protease inhibitors for 20 min on ice to resuspend the nuclei pellet. The suspension was collected as the NPC fraction, diluted eightfold in PBS containing protease inhibitors, sonicated on ice, and cleared by centrifugation at 1000 × g for 10 min.

## Fluorescence microscopy and analysis

Immunofluorescence was conducted as previously described (*Wang et al., 2015*). Briefly, conjugates of Jurkat T cells and Raji APCs were plated on poly-L-lysine-coated slides, incubated for 15 min at room temperature, fixed for 15 min with 4% PFA, and permeabilized with 0.2% Triton X-100 for 10 min at room temperature. The slides were blocked with 2% BSA for 1 hr, and samples were stained with indicated antibodies overnight at 4°C. After washing with PBS, slides incubated for 1 hr at room temperature with secondary antibodies. After three washes with PBS, the cells were mounted with a drop of mounting medium. Images were obtained with a Leica SP5 laser-scanning confocal microscope equipped with 100× objective lens with laser excitation at 405 nm, 488 nm, 561 nm, or 633 nm. Each image is a single z-section, and the z-position closest to the center of the cell (the equatorial plane) was chosen. Images were analyzed and processed with ImageJ and Adobe Photoshop CS6 software. Briefly, quantitative colocalization analysis of confocal microscopy images was performed with the JACoP module of the FIJI-ImageJ software (https://imagej.nih.gov/ij/index.html) to generate the Pearson's correlation coefficient (PCC) with a range of 1 (perfect correlation) to −1 (perfect exclusion). PCC measures the pixel-by-pixel covariance in the signal levels of two channels of an image. The protein nuclear/cytoplasmic (N/C) expression ratio of confocal microscope images was quantified as follows: N/C = Total fluorescence intensity in nuclear area/(Total fluorescence intensity in whole-cell area – total fluorescence intensity in nucleus). The fluorescence intensities of PKC-θ or RanGAP1 in the NE were quantified with FIJI-ImageJ software. The region of NE was automatically segmented as described with FIJI-ImageJ software (*Tosi et al., 2020*).

## Expression and purification of GST-fusion proteins

GST-fusion proteins were expressed in *E. coli* BL21 after induction with 0.3 mM isopropyl-beta-D-thiogalactopyranoside (Sangon Biotech) for 12 hr at 18°C. Bacteria were resuspended in lysis buffer (1× PBS: 137 mM NaCl, 2.7 mM KCl, 10 mM Na$_2$HPO$_4$, and 1.8 mM KH$_2$PO$_4$, pH 7.4; proteinase

inhibitors, and 1% Triton X-100 for GST-Nups or 0.1% Triton X-100 for GST-RanGAP1). Bacterial extracts were sonicated for 10 min and centrifuged. GST-fusion proteins were purified by incubation with glutathione-Sepharose beads (GE Healthcare). The precipitates were washed 3× with lysis buffer, then eluted with elution buffer (50 mM Tris–HCL, pH 8.0, 15 mM reduced glutathione). Coomassie Brilliant Blue staining was used as loading control.

### PKC-$\theta$ in vitro kinase assay

The kinase assay was conducted as previously described (*Wang et al., 2015*). Jurkat T-cell lysates were immunoprecipitated with anti-PKC-θ or a control IgG. The IPs were washed 5× with RIPA buffer containing 0.2% SDS and 1× with PKC-θ kinase buffer (20 mM HEPES, pH 7.2–7.4, 10 mM MgCl$_2$, and 0.1 mM EGTA) and were resuspended in 25 µl of kinase buffer containing 20 µM cold ATP, 10 µM PMA, 200 µg/ml phosphatidyserine, and 1 µg of recombinant GST-RanGAP1 protein or GST-RanGAP1 mutant proteins as substrate. After incubation for 30 min at 30℃ with gentle shaking, the reaction was stopped by adding 5× loading buffer. Samples were boiled at 95℃ for 10 min, separated by SDS–PAGE, and detected by western blot with an anti-p-Ser/Thr antibody (a mixture of phospho-Ser- and phospho-Thr-specific antibodies, BD Biosciences, 9631).

### Mass spectrometric analysis

Samples for co-IP were prepared as described previously with minor modification (*Wang et al., 2015*). In brief, anti-CD3/CD28-stimulated Jurkat T cells were lysed and followed by IP with anti-PKC-θ or anti-IgG. IPs immobilized on protein G beads and 1 µg recombinant GST-RanGAP1 protein immobilized on GSH Sepharose beads were separately washed 2× with kinase buffer and mixed to initiate the kinase assay. Reactions were terminated by Laemmli sample buffer, boiled, and resolved on SDS–PAGE. Gel bands of interest were excised and subjected to tryptic digestion. After desalting, the peptides were analyzed by tandem MS. A splitless Ultra 2D Plus system (Eksigent) coupled to the TripleTOF 5600 System (AB SCIEX) with a Nanospray III source (AB SCIEX) were performed to analyze immunoprecipitated proteins and identify posttranscriptional modification sites of RanGAP1. The search engine ProteinPilot V4.5 was used to assigned potential modification sites with high confidence.

### Immunoelectron microscopy

Immunoelectron microscopy was performed on Jurkat T cells stimulated for 0–15 min with anti-CD3 plus anti-CD28. Cells were fixed in buffer (4% paraformaldehyde, 0.2% glutaraldehyde), pelleted, treated with LR white acrylic resin (L9774, Sigma-Aldrich), and then frozen for 72 hr at −20℃ with UV irradiation. Frozen pellets were sectioned by a cryo-ultramicrotome (EM UC6 and FC6, Leica). Cryosections were thawed, rinsed in PBS with 1% glycine, and incubated in 0.01 M PBS containing 0.1% BSA and 5% goat serum for 30 min at room temperature. The samples were incubated with mouse anti-PKC-θ antibody diluted 1:50 overnight at 4℃, and rinsed in 0.01 M PBS, then incubated with 10 nm colloidal gold-labeled anti-mouse IgG secondary antibody (G7652, Sigma-Aldrich) diluted 1:25 for 3 hr at room temperature. Grids were then rinsed in 0.01 M PBS and ultrapure water and embedded in 2% uranyl acetate with lead citrate. Images were taken on an electron microscope (JEM1400, JEOL).

### Enzyme-linked immunosorbent assay

Samples for enzyme-linked immunosorbent assay (ELISA) were prepared as described previously (*Wang et al., 2015*). Aliquots of T cells ($3 \times 10^6$) transfected with HA-RanGAP1 or HA-RanGAP1$^{AA}$ vectors were stimulated for 24 hr with anti-CD3 plus anti-CD28, and the concentration of IL-2 in culture supernatants was determined by ELISA according to the manufacturer's instructions (BD Biosciences).

### CRISPR/Cas9 gene editing

LentiCRISPRv2 which contains Cas9 and GFP was used to edit a RanGAP1 genomic fragment in the Jurkat E6.1 cell line. The gRNA targeting sequences are listed in the Key resources table. Cells in a logarithmic growth phase were transfected with plasmids encoding gRNA by nucleofection (Lonza 4D Nucleofector system). Forty-eight hours after transfection, GFP-positive cells were sorted on a

flow cytometer (BD FACSAria II) and seeded individually into 96-well plates. The RanGAP1genome-edited cell line was identified by genomic DNA sequencing.

## Retrovirus transduction

Platinum-E packaging cells were plated in a six-well plate in 2 ml RPMI-1640 medium plus 10% FBS. After 24 hr, cells were transfected with empty pMX vector or RanGAP1 (AA or EE)-expressing vector DNA (5 μg) with Lipofectamine 3000 Reagent (Thermo Fisher Scientific). After an overnight incubation, the medium was replaced and cultures were maintained for another 24 hr. Retroviral supernatants were then collected and filtered, supplemented with 5 μg/ml of polybrene and 100 U/ml of recombinant IL-2, and then used to infect T cells that had been pre-activated for 48 hr with plate-bound monoclonal anti-CD3 antibody (5 μg/ml) and soluble anti-CD28 monoclonal antibody (2.5 μg/ml) in the presence of recombinant IL-2 (100 U/ml). Plates were centrifuged for 1 hr at 2000 r.p.m. and incubated for 4 hr at 32°C and overnight at 37°C, followed by one additional retroviral transduction the next day. On day 4, cells were washed and cultured in RPMI-1640 medium containing 10% (vol/vol) FBS and recombinant IL-2 (100 U/ml) for another 3 days before re-stimulation with monoclonal anti-CD3 plus –CD28 antibodies.

## Computational analysis

I-Mutant2.0 (folding.biofold.org/i-mutant/i-mutant2.0.html) was using to predict RanGAP1 protein stability changes with single point mutation from the protein sequence (NP_002874.1).

## Statistical analysis

Prism (GraphPad 6.0 Software) was used for graphs and statistical analysis. Statistical analysis was performed with a two-tailed, unpaired Student's t-test or one-way ANOVA with post hoc test. p-values of less than 0.05 were considered statistically significant. Graphs represent mean ± standard error of the mean (s.e.m).

## Acknowledgements

Supported by the National Natural Science Foundation of China (31670893, 31370886, 31170846), the Guangzhou Science and Technology Project (201904010445), Guangdong Provincial Natural Science Foundation (2021A1515010543), and Guangdong Science and Technology Department (2020B1212060031). We thank Dr. G Fu for helpful discussion and Dr. M Zhao for help in PBMC-related experiment.

## Additional information

### Funding

| Funder | Grant reference number | Author |
|---|---|---|
| National Natural Science Foundation of China | 31670893 | Yingqiu Li |
| National Natural Science Foundation of China | 31370886 | Yingqiu Li |
| National Natural Science Foundation of China | 31170846 | Yingqiu Li |
| Guangzhou Science and Technology Program key projects | 201904010445 | Yingqiu Li |
| Guangdong Basic and Applied Basic Research Foundation | 2021A1515010543 | Yingqiu Li |
| Guangdong Science and Technology Department | 2020B1212060031 | Yingqiu Li |

The funders had no role in study design, data collection and interpretation, or the decision to submit the work for publication.

## Author contributions
Yujiao He, Data curation, Software, Formal analysis, Validation, Investigation, Visualization, Methodology, Writing - original draft; Zhiguo Yang, Yu Gong, Yun-Yi Li, Yiqi Chen, Yunting Du, Dianying Feng, Investigation; Chen-si Zhao, Formal analysis, Validation, Investigation, Visualization, Methodology; Zhihui Xiao, Formal analysis, Validation, Investigation; Amnon Altman, Conceptualization, Resources, Writing - review and editing; Yingqiu Li, Conceptualization, Resources, Data curation, Formal analysis, Supervision, Funding acquisition, Investigation, Visualization, Methodology, Writing - original draft, Project administration, Writing - review and editing

## Author ORCIDs
Yingqiu Li (iD) https://orcid.org/0000-0003-3206-5231

## Ethics
Animal experimentation: This study was performed in strict accordance with the recommendations in the Guide for the Care and Use of Laboratory Animals of the Sun Yat-Sen University. All of the animals were handled according to guidelines approved by the Animal Care and Ethics committee of Sun Yat-Sen University. The protocol was approved by the Committee on the Ethics of Animal Experiments of Sun Yat-Sen University (Permit Number: SYSU-IACUC-2019-B616). The mice were euthanized by $CO2$ from compressed gas cylinders, and we complied with all the ethical regulation.

## Decision letter and Author response
Decision letter https://doi.org/10.7554/eLife.67123.sa1
Author response https://doi.org/10.7554/eLife.67123.sa2

# Additional files

## Supplementary files
• Transparent reporting form

## Data availability
All data generated or analysed during this study are included in the manuscript and supporting files.

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
