## [Decision Letter]

**Acceptance summary:**

This paper is of interest for immunologists studying antigen receptor signaling and scientists interested in nuclear import. The work provides a novel molecular mechanism linking antigen receptor signalling to an indirect nuclear transport mechanism for regulating localisation of multiple key transcription factors for the T cell immune response during early activation.

**Decision letter after peer review:**

Thank you for submitting your article "TCR signaling promotes the assembly of RanBP2/RanGAP1-SUMO1/Ubc9 NPC subcomplex via phosphorylation of RanGAP1 by PKC-θ" for consideration by *eLife*. Your article has been reviewed by 3 peer reviewers, including JC Zúñiga-Pflücker as the Reviewing Editor and Reviewer #1, and the evaluation has been overseen by Jonathan Cooper as the Senior Editor.

Essential revisions:

1) A common concern was the need to show whether forced expression of activated RanGAP1 could rescue NPC function in PKC-θ KO T cells, which would strengthen the manuscript and the conclusion that the RanBP2 complex formation is the key step downstream of PKC-θ signalling that is regulating nuclear transport of transcription factors.

2) The authors should address whether the PKC-θ defect in nuclear transport is a generalized effect on many proteins/RNA beyond the transcription factors that were measured and reported.

3) Limiting the conclusions to the timeframe analyzed would be more appropriate.

*Reviewer #1 (Recommendations for the authors):*

1) The authors show ERK phosphorylation in most of their whole cell lysate preparations (e.g., Figure 1i and others), likely to indicate that the cells used for the lysates had in fact undergone activation. However, it is interesting that in PKC-θ deficient (Prkcq-/-) samples there appears to be a significant reduction in the levels of phosho-ERK, was this expected, or was this due to unanticipated regulation by PKC-θ of ERK activation or is this due to some other feedback event. The authors should discuss their interpretation of this phenomenon.

2) The authors point out that the results shown in Figure 3D, 3F, looking at the role of PKC-θ in thymocytes are consistent with previous results showing that thymocytes did not require PKC-θ for their maturation (PMID: 10746729); however, later studies (PMID: 18802072) showed that PKC-θ can play a role an important role in thymocyte selection. This role for PKC-θ should be discussed, in particular in light of the lowered ERK-phosphorylation seen in the thymocyte samples from the Prkcq-/- mice. Also, this begs the question as to how the NPC is regulated and able to bring in AP1, NFkB and NFAT in thymocytes without PKC-θ.

3) The legend for Figure 6 should mention that the cells being analyzed are control and gene-modified Jurkat cells.

*Reviewer #2 (Recommendations for the authors):*

PKC-θ is a critical signaling molecule downstream of T cell receptor (TCR), and required for T cell activation via regulating the activation of transcription factors including AP-1, NF-κB and NFAT. This manuscript revealed a novel function of PKC-θ in the regulation of the nuclear translocation of these transcription factors via nuclear pore complexes. This novel perspective for PKC-θ function advances our understanding T cell activation. Although the manuscript provided many evidence to support the conclusion, however there are some concerns:

1. Importin-β has minimum binding to NPC in PKC-θ KO T cells, does this means that there is almost no importin-β-dependent transportation in PKC-θ KO which is alive and many nuclear proteins and cytoplasm mRNA need to be transported. Is there a way to measure overall importin-β-mediated nucleo-cytoplasmic transportation?

2. Does importin-β mediates transportation both ways, nucleus to cytoplasmic, and cytoplasmic to nucleus? If that is the case, does the two-way transportation are all affected by PKC-θ. There should be a way to monitor importin-β-dependent transportation each way.

3. It is mentioned that PKC-θ may promote c-Jun nuclear import, but not its phosphorylation per se, suggesting that it regulates AP-1 activation primarily by controlling the function of the NPC. This is a very interesting suggestion. To support this possibility, first, the c-Jun phosphorylation in PKC-θ KO T cells should be measured. In addition, does constitutively active JNK phosphorylate and activate c-Jun in PKC-θ KO T cells?

4. The authors should compare RanGAP1AA and RanGAP1EE in the regulation of AP-1, NFAT and NF-κB nuclear translocation in WT and PKC-θ KO T cells. In addition, as controls, they should monitor transport of other nuclear proteins, particularly those constitutively transported. It is necessary to have more evidence that PKC-θ regulates NPC import specifically, and not generally affect NPC function so as to inhibit cell survival.

*Reviewer #3 (Recommendations for the authors):*

1) At points the methods section is lacking in detail – the manuscript figures should be checked to verify that all procedures used are adequately described in the methods section – for example, it is not specified how plasma membrane was purified for Figure 1A-B (best guess from methods is that this is the cytoplasmic fraction? – this needs clarification either way); there is no detail about how transfections were performed beyond the statement cells were "electroporated", and no reference to how CRISPR/Cas9 editing of Jurkat cell line to generate RanGAP1 knockdown cells was performed (this is only mentioned in the figure legend for Figure S6 with no further details).

2) In Figure 2F legend it says that PKC-θ mutant K325R/K506R is one of the mutants used in this experiment however there is not data for this shown in the figure.

3) The AA, EE abbreviations used in the figures 4-6 were only defined in the main text. If this abbreviation could be included in the figure legends as well this would make figures easier to examine and interpret.

---

## [Author Response]

Essential revisions:1) A common concern was the need to show whether forced expression of activated RanGAP1 could rescue NPC function in PKC-θ KO T cells, which would strengthen the manuscript and the conclusion that the RanBP2 complex formation is the key step downstream of PKC-θ signalling that is regulating nuclear transport of transcription factors.

As suggested, we transfected PKCθ-knockdown Jurkat T cells or transduced murine *Prkcq*^-/-^ splenic T cells with a constitutively active RanGAP1-EE mutant or negative control RanGAP1-AA mutant expression vectors and analyzed the rescue of NPC function. Forced expression of RanGAP1-EE rescued the nuclear import of c-Jun and importin-β, but not of NFATc1 or p65. The inability to rescue the nuclear translocation of NFATc1 and p65 is expected because their upstream signaling pathways are known to be disrupted in *Prkcq*^-/-^ T cells (Pfeifhofer et al., 2003; Sun et al., 2000). Thus, we conclude that RanBP2 subcomplex formation is the key step downstream of PKCθ signaling, which regulates NPC function, including the nuclear import of c-Jun. The results of this experiment (new Figure 8) are highlighted in in the main text, p. 13.

2) The authors should address whether the PKC-θ defect in nuclear transport is a generalized effect on many proteins/RNA beyond the transcription factors that were measured and reported.

To address this question, we repeated the subcellular fractionation experiment and compared the nuclear translocation of additional proteins in wild-type and PKCθ-knockdown Jurkat T cells (new Figure 7). We found that PKCθ-RanGAP signaling axis is differentially required in different nuclear transport pathways. Specifically, we tested importin-β1-dependent proteins, including RNA exosome complex subunit Dis3, ribosome subunits RPS3 and RPL26, tumor suppressor p53, histone H1; Ran-dependent but importin-β1-independent proteins, including histones (H2B, H3) and protein/RNA export receptor CRM1; and the mRNA export receptor NXF1 which is Ran-independent (Bernardes and Chook, 2020; Serpeloni, Vidal, Goldenberg, Avila, and Hoffmann, 2011).

We found that following PKCθ knockdown in resting T cells, the nuclear import of Dis3, RPS3 and RPL26, and the nuclear export of CRM1 were decreased, consistent with their requirement for importin β1 or the Ran system; however, the Ran-independent NXF1 also displayed nuclear retention. In TCR-activated cells, PKCθ knockdown did not affect the nuclear transport of CRM1, Dis3, RPS3, or RPL26, implying that under activation conditions, other signaling pathways may overcome the effect of PKCθ deficiency on nuclear transport (such as Dis3) or result in a phenotype similar to that of siPKCθ (such as CRM1). However, TCR stimulation did not alleviate the siPKCθ-induced defects on importin-β1 nuclear import or NXF1 nuclear export. Interestingly, independent of the stimulation status, PKCθ knockdown did not affect the nuclear transport of histones. Moreover, the nuclear import of stimulation-induced p53 was also not affected by siPKCθ. In summary, neither PKCθ nor TCR signaling regulates histones translocation; in resting cells, the importin-β1- or Ran-dependent nuclear transport pathways we tested, with the exception of histones, are dependent on PKCθ; in activated cells, PKCθ is dispensable for their nuclear transport, except for the nuclear transport of importin-β1; and PKC-θ is required for NXF1 transport under both statuses. Thus, the PKCθ-RanGAP signaling axis is differential required in different nuclear transport pathways. These results are described and discussed in the main text, p. 12-13, p. 15.

3) Limiting the conclusions to the timeframe analyzed would be more appropriate.

We addressed this comment by repeating the experiments analyzing RanGAP1 binding to NPCs and of c-Jun nuclear translocation in PKCθ-deficient cells following TCR stimulation for a longer time (12 hours). The results were similar to those obtained after a 15-minute stimulation, namely, PKCθ deficiency led to a reduction of RanGAP1 binding to NPC and c-Jun nuclear translocation, suggesting that there is no compensation even after a 12-hour stimulation period. This result validates our original conclusion. The results are shown in Figure 3—figure supplement 1G, H, and are described in detail in the main text, p. 9.*Reviewer #1 (Recommendations for the authors):*

1) The authors show ERK phosphorylation in most of their whole cell lysate preparations (e.g., Figure 1i and others), likely to indicate that the cells used for the lysates had in fact undergone activation. However, it is interesting that in PKC-θ deficient (Prkcq-/-) samples there appears to be a significant reduction in the levels of phosho-ERK, was this expected, or was this due to unanticipated regulation by PKC-θ of ERK activation or is this due to some other feedback event. The authors should discuss their interpretation of this phenomenon.

The defective TCR-induced ERK activation in *Prkcq*^-/-^ T cells was expected since it has been reported that, upon TCR stimulation, PKCθ activates ERK via phosphorylating RasGRP1 to drive Ras-Erk activation (Mol Cell Biol. 2005, 25:4426-4441.). We have added this information and the discussion and highlighted in the main text, p. 8, p. 16-17.

2) The authors point out that the results shown in Figure 3D, 3F, looking at the role of PKC-θ in thymocytes are consistent with previous results showing that thymocytes did not require PKC-θ for their maturation (PMID: 10746729); however, later studies (PMID: 18802072) showed that PKC-θ can play a role an important role in thymocyte selection. This role for PKC-θ should be discussed, in particular in light of the lowered ERK-phosphorylation seen in the thymocyte samples from the Prkcq-/- mice. Also, this begs the question as to how the NPC is regulated and able to bring in AP1, NFkB and NFAT in thymocytes without PKC-θ.

As suggested, we have now discussed the role for PKCθ in thymocyte selection in more detail (p. 16-17). While the defects of NPC assembly and of AP-1, NF-κB and NFAT activation could be compensated by other PKC isoforms, such as PKCη (Figure 3F, J; Altman and Kong, 2016; Fu et al., 2011; Morley et al., 2008; Sun et al., 2000), the non-redundant role of PKCθ in TCR-induced ERK activation reveals that PKCθ is critical for positive selection as well as for the activation of mature T cells (Altman and Kong, 2016; Morley et al., 2008). Indeed, a personal communication with Dr. G. Fu who reported the cooperation between PKCη and PKCθ in thymic selection (Fu et al., Sci Signal., 2011), confirmed that PKCη deficiency had no impact on ERK activation in thymocytes, although it did impair TCR-induced NF-κB activation and Ca^2+^ signaling.

3) The legend for Figure 6 should mention that the cells being analyzed are control and gene-modified Jurkat cells.

The requested information has been added in the Figure 6 legend.

Reviewer #2 (Recommendations for the authors):PKC-θ is a critical signaling molecule downstream of T cell receptor (TCR), and required for T cell activation via regulating the activation of transcription factors including AP-1, NF-κB and NFAT. This manuscript revealed a novel function of PKC-θ in the regulation of the nuclear translocation of these transcription factors via nuclear pore complexes. This novel perspective for PKC-θ function advances our understanding T cell activation. Although the manuscript provided many evidence to support the conclusion, however there are some concerns:1. Importin-β has minimum binding to NPC in PKC-θ KO T cells, does this means that there is almost no importin-β-dependent transportation in PKC-θ KO which is alive and many nuclear proteins and cytoplasm mRNA need to be transported. Is there a way to measure overall importin-β-mediated nucleo-cytoplasmic transportation?

Please see our response to comment #2 in the Essential revisions section above.

2. Does importin-β mediates transportation both ways, nucleus to cytoplasmic, and cytoplasmic to nucleus? If that is the case, does the two way transportation are all affected by PKC-θ. There should be a way to monitor importin-β-dependent transportation each way.

We addressed this question by analyzing the effect of PKCθ deficiency on the fate of Ran-dependent protein/RNA export receptor (CRM1), both Ran- and importin-β-dependent cargos, namely, ribosome subunits, RNA exosome subunit Dis3, histone H1, etc., and Ran-independent mRNA export receptor, NXF1. The results are shown in new Figure 7 and described in p. 12-13, p. 15.

3. It is mentioned that PKC-θ may promote c-Jun nuclear import, but not its phosphorylation per se, suggesting that it regulates AP-1 activation primarily by controlling the function of the NPC. This is a very interesting suggestion. To support this possibility, first, the c-Jun phosphorylation in PKC-θ KO T cells should be measured. In addition, does constitutively active JNK phosphorylate and activate c-Jun in PKC-θ KO T cells?

We analyzed the phosphorylation of JNK and c-Jun in PKC-θ KO T cells and found that PKC-θ deficiency did not impair their activation (Figure 3—figure supplement 1C-D).

4. The authors should compare RanGAP1AA and RanGAP1EE in the regulation of AP-1, NFAT and NF-κB nuclear translocation in WT and PKC-θ KO T cells. In addition, as controls, they should monitor transport of other nuclear proteins, particularly those constitutively transported. It is necessary to have more evidence that PKC-θ regulates NPC import specifically, and not generally affect NPC function so as to inhibit cell survival.

As suggested, we analyzed the effect of RanGAP1 AA and EE mutants on AP-1, NFAT and NF-κB nuclear translocation in WT and PKCθ KO or KD T cells (new Figure 8). In WT T cells, RanGAP1AA and EE did not affect the translocation of these transcription factors since the NPC is already occupied by endogenous RanGAP1-SUMO1. In PKCθ-deficient cells, RanGAP1EE, but not RanGAP1AA, could rescue c-Jun’s nuclear import. None of these mutants rescued the nuclear translocation of NFAT and NF-κB because, as discussed above, they are regulated by upstream, TCR-proximal and PKCθ-dependent signals.

Please also see our response above to comment #2 in the Essential revision section, and the new Figure 7.

Reviewer #3 (Recommendations for the authors):1) At points the methods section is lacking in detail – the manuscript figures should be checked to verify that all procedures used are adequately described in the methods section – for example, it is not specified how plasma membrane was purified for Figure 1A-B (best guess from methods is that this is the cytoplasmic fraction? – this needs clarification either way); there is no detail about how transfections were performed beyond the statement cells were "electroporated", and no reference to how CRISPR/Cas9 editing of Jurkat cell line to generate RanGAP1 knockdown cells was performed (this is only mentioned in the figure legend for Figure S6 with no further details).

We apologize for this omission. We have now added the protocol of plasma membrane purification for Figure 1A, B in the Materials and methods section on p. 23; the electroporation information on p. 22; the protocol of CRISPR/Cas9 editing on p. 26.

2) In Figure 2F legend it says that PKC-θ mutant K325R/K506R is one of the mutants used in this experiment however there is not data for this shown in the figure.

We apologize for incorrectly labelling the figure. The PKCθ-2KR refers to the K325R/K506R mutant. We have added this information in the Figure 2F legend.

3) The AA, EE abbreviations used in the figures 4-6 were only defined in the main text. If this abbreviation could be included in the figure legends as well this would make figures easier to examine and interpret.

As suggested, the AA, EE abbreviations are now defined in the figure legends for Figures 4-6.